# Non-surgical treatments for post-burn scars: A network meta-analysis

**Xiaojuan Yang**[1,2☯], **Rong Li**[2,3☯], **Xiaorong Mao**[2,3], **Shuangying Gong**[2], **Yu Fan**[2], **Chunyi Xu**[1], **Qing Wen**[2,3], **Xiaotao Xu**[2,3], **Wei Li**[1,2]*

**1** Department of Burn and Wound Repair Surgery, Sichuan Provincial People's Hospital, Chengdu, Sichuan, China, **2** School of Medicine, University of Electronic Science and Technology of China, Chengdu, Sichuan, China, **3** Department of Nursing, Sichuan Provincial People's Hospital, Chengdu, Sichuan, China

☯ These authors contributed equally to this work, sharing first authorship on this work.
* 2220147@uestc.edu.cn

## Abstract

### Background and aim

Post-burn scarring is a prevalent condition, and the existing non-surgical treatments exhibit varying degrees of efficacy. There is limited evidence available to determine the effective non-surgical treatment for post-burn scars. This study employs a multi-index network meta-analysis to conduct a comprehensive evaluation and comparative ranking of non-surgical treatments for post-burn scars. The aim is to identify the most effective treatment methods, thereby providing a robust, evidence-based foundation to guide clinical decision-making.

### Methods

PubMed, Web of Science, Cochrane Library, PEDro, and Embase were systematically searched for eligible randomized controlled trial studies, and the network meta-analysis was performed via a frequentist approach. The primary outcomes assessed were Vancouver Scar Scale score, scar thickness and Visual Analogue Scale score.

### Results

A total of 17 studies and 1,013 participants were included in this analysis. The treatment ranking revealed that massage therapy demonstrated the most significant efficacy in reducing Vancouver Scar Scale score (surface under the cumulative ranking curve [SUCRA] = 89.0%), $CO_2$ laser therapy exhibited the highest efficacy in decreasing scar thickness (SUCRA = 96.8%), and extracorporeal shock wave therapy + routine treatment showed the most significant efficacy in reducing Visual Analogue Scale score (SUCRA = 58.6%).

**Data availability statement:** All relevant data are within the manuscript and its Supporting Information files.

**Funding:** This work was supported by the Sichuan Province Science and Technology Program (Grant No：23ZDYF1907), awarded to Dr. Wei Li, who served as the principal investigator. The funders had no role in study design, data collection and analysis, decision to publish, or preparation of the manuscript.

**Competing interests:** The authors declare no competing interests.

**Abbreviations:** VSS, Vancouver Scar Scale; VAS, Visual Analogue Scale; PT, pressure therapy; MT, massage therapy; SG, silicone gel; ESWT, extracorporeal shock wave therapy; LT, laser therapy; $CO_2LT$, $CO_2$ laser therapy; PTS, pressure therapy and silicone; OPLT, orange polarized light therapy; HILT, high intensity laser therapy; PDL, pulsed dye laser; RT, routine treatment; PRISMA, Preferred Reporting Items for Systematic Reviews and Meta-analysis; RCTs, randomized controlled trials; SD, standard deviation; CI, confidence interval; SMD, standard mean difference.

## Conclusion

This network meta-analysis illustrates that massage therapy, $CO_2$ laser therapy and extracorporeal shock wave therapy + routine treatment are the most effective non-surgical treatments for reducing Vancouver Scar Scale score, scar thickness and Visual Analogue Scale score for post-burn scars, respectively. However, the findings reflect outcomes at a specific stage of scar maturation. And our conclusions must be interpreted with caution due to the limited number of studies included. In the future, well-designed randomized controlled trials with a large sample size are needed to validate these findings.

## 1 Introduction

Burn injuries present a significant global healthcare challenge, with 30% to 90% of survivors developing pathological scarring, predominantly in the form of keloids and hypertrophic scars. These conditions disrupt the intricate process of physiological wound repair [1]. Such aberrant healing responses not only lead to chronic pain, persistent itching, and joint contractures but also severely impact psychosocial well-being and functional capacity, thereby significantly reducing quality of life [2]. Effective scar management emerges as a critical determinant of recovery outcomes in burn patients [3], particularly within the first year after injury, when scars are immature and more responsive to treatment. Currently, non-surgical treatments are often the initial treatment of choice for inhibiting or slowing scar progression [4–6].

Silicone products, pressure therapy (PT), massage therapy (MT), and extracorporeal shock wave therapy (ESWT) are widely recognized non-surgical treatments for post-burn scars [7]. These interventions have demonstrated significant efficacy in improving both the symptoms and appearance of post-burn scars, while also being well-tolerated by patients [8–10]. In addition to the aforementioned non-surgical treatments, CO2 laser therapy ($CO_2LT$) and microneedling can be considered important components of non-surgical treatment strategies for post-burn scars [11]. $CO_2LT$ is used to resurface the skin, promoting collagen remodeling and improving scar texture and appearance. Though it involves a higher degree of invasiveness compared to silicone products or PT, it does not typically require general anesthesia, especially when applied in fractional modes. Microneedling involves creating micro-injuries to stimulate collagen production and improve scar healing. Although it requires penetration of the skin, microneedling can be performed under local anesthesia. Despite their relatively higher invasiveness levels compared to other modalities mentioned earlier, $CO_2LT$ and microneedling do not involve surgical tissue removal, thus they remain classified as non-surgical treatments.

Non-surgical techniques offer several key advantages that make them highly relevant in modern burn care. Firstly, these methods are minimally invasive or non-invasive, which reduces the risk of complications and enhances patient safety. Secondly, compared to surgical options, they are often easier to implement, making them accessible in various clinical settings. Thirdly, these treatments can significantly

improve patients' quality of life by alleviating pain, reducing scar stiffness, and enhancing cosmetic outcomes. By focusing on non-surgical interventions, healthcare providers can offer effective solutions that align with patient preferences for less invasive treatments, while also minimizing recovery time and improving resource utilization. The growing adoption of these therapies underscores their potential to influence comprehensive, patient-centered burn care strategies [12].

Evaluating the efficacy of treatments for post-burn scars is crucial for guiding clinical decision-making and enhancing therapeutic outcomes. To achieve this, various assessment tools have been developed to evaluate different aspects of scar characteristics. The Vancouver Scar Scale (VSS) is one of the most commonly used tools for assessing scar characteristics, including vascularity, pliability, pigmentation, and height [13]. Although VSS has undergone extensive review and remains widely utilized, available data on its reliability have received indeterminate quality ratings [14]. Furthermore, VSS may lack sensitivity to subtle changes in scar appearance, and its subjective components can lead to variability in scoring [15]. Despite these limitations, studies have shown that VSS demonstrates moderate inter-rater reliability when applied by trained evaluators [16]. The Visual Analog Scale (VAS) is another essential tool, primarily used for assessing pain and discomfort associated with scars. It quantifies a patient's subjective experience by asking them to indicate their pain level on a linear scale. The simplicity and ease of use of VAS make it a popular choice for pain assessment [17]. However, VAS scores are heavily reliant on patient self-reporting, which can be influenced by individual biases or external factors. Scar thickness serves as an objective indicator that can be accurately measured using ultrasound or other imaging techniques [18]. Monitoring scar thickness enables clinicians to adjust treatmsent regimens, such as increasing the frequency of interventions or modifying treatment methods, thereby optimizing therapeutic effectiveness. Collectively, the VSS captures morphological and physiological scar characteristics, the VAS quantifies patient-reported symptomatic experiences, and scar thickness measurements offer objective structural insights. These metrics address the multidimensional nature of post-burn scars evaluation, enabling comprehensive analysis of non-surgical treatment impacts.

Finnerty et al. pointed out that it was essential to confirm the effectiveness of both existing and new therapies for post-burn scars [19]. In recent years, several systematic reviews have been performed to validate the efficacy of non-surgical treatments for post-burn scars. Zuccaro et al. reported that they were unable to draw definitive conclusions regarding the effectiveness of laser therapy for hypertrophic burn scars [20]. Aultet al. and Lin et al. reported that MT significantly improved scar formation [21,22]. Aguilera-Saez et al. suggested that scientific evidence on the use of ESWT for the treatment of burn patients was weak due to the lack of studies and their low quality, but ESWT is a promising tool in this field [23]. Based on low-to-moderate quality evidence, Santuzzi et al. concluded that massage, laser and shock wave therapy could reduce pain, whereas massage, shock wave therapy and silicone had negligible or unclear effects on improving scar elasticity and vascularisation [10]. Although previous systematic reviews have provided valuable insights into non-surgical treatments for post-burn scars, a clear understanding of their specific effectiveness remains elusive. Several key limitations in prior studies contribute to this gap. First, many studies utilized different methodologies, outcome measures, and follow-up durations, complicating efforts to draw definitive conclusions about the comparative efficacy of various treatments. Second, several earlier studies were limited by small sample sizes, which reduced their statistical power and generalizability. Finally, the inconsistent use of assessment tools across studies introduced variability in how outcomes were measured and reported, further hindering meaningful comparisons.

The network meta-analysis (NMA) enables the simultaneous comparison of multiple interventions by synthesizing data from various studies, thereby offering clearer and more robust conclusions regarding treatment effectiveness. This study employs NMA to provide a comprehensive evaluation of non-surgical treatments for post-burn scars. By focusing on key outcomes such as VSS score, scar thickness measurements, and VAS score, aiming to assist healthcare professionals and patients in selecting the most effective non-surgical interventions to improve post-burn scars.

## 2 Methods

### 2.1 Study protocol registration

The study was conducted in accordance with the Preferred Reporting Items for Systematic Reviews and Meta-analysis (PRISMA) guidelines. The protocol for this study was registered in the International Prospective Register of Systematic Reviews (PROSPERO) in April 2024 (CRD42024532528).

### 2.2 Search strategy

PubMed, Web of Science, Cochrane Library, PEDro, and Embase were systematically searched from their inception to 5 May 2024 for eligible studies (S2 File). The search terms were as follows: (burns OR burn OR thermal injury) AND (cicatrix OR cicatrization OR scars OR scar*) AND (random allocation OR randomized controlled trial OR randomised controlled trial OR random* OR randomized OR randomised OR RCT). Additionally, the reference lists of the included studies and relevant systematic reviews were manually searched to identify any additional eligible studies.

### 2.3 Inclusion and exclusion criteria

Articles were included if (1) they were published in peer-reviewed journals; (2) they reported non-surgical treatments for patients with post-burn scars, including monotherapies or combination therapies; (3) they reported the following outcomes at approximately 6 months post-injury: VSS score, scar thickness measured by ultrasound, and pain intensity assessed via VAS score; (4) they were randomized controlled trials (RCTs). In this NMA, $CO_2LT$ refers specifically to the ablative procedure.

Articles were excluded if (1) they were clinical trials, review articles, letters to the editor or case series; (2) they were not published in English; (3) they reported incomplete data; (4) the full text was unavailable. Additionally, conference abstracts were excluded due to insufficient detail on study design, interventions, and outcomes, which would limit the ability to accurately assess risk of bias and interpret treatment effects.

### 2.4 Study selection

Two researchers (CYX & QW) independently screened the retrieved literature. Firstly, duplicate records were removed using EndNote software (version 20.0). Subsequently, articles were assessed for inclusion and exclusion based on the title, abstract and full text. Finally, the screening results were verified, with any disagreements being resolved by discussion.

### 2.5 Data extraction

Two authors (SYG & YF) independently extracted the data using a structured format. The extracted information included: the name of the first author, publication year, country, sample size (for each arm), mean age or range, duration from injury, scar location, intervention methods and reported outcomes [mean±standard deviation (SD)]. In this study, the primary outcomes were VSS score, scar thickness and VAS score. For outcomes reported at multiple time points, only data closest to 6 months were extracted. If mean differences and standard deviation (SD) were reported for each outcome, the data were extracted directly. Or these values were calculated based on pre- and post-treatment mean±SD, sample sizes, standard error, confidence interval (CI) and p-value. The calculated formulas were from the Cochrane Manual (https://training.cochrane.org/handbook/current/chapter-06). Any disagreements were resolved by discussion or consensus.

### 2.6 Risk of bias

Two independent reviewers (XRM and XTX) assessed the risk of bias for all included studies using the Cochrane Risk of Bias Tool [24], included: (1) bias arising from the randomization process; (2) bias due to deviations from intended

interventions; (3) bias due to missing outcome data; (4) bias in measurement of the outcome; (5) bias in selection of the reported result. Each domain was categorized as " low risk of bias ", " some concerns " or " high risk of bias ". When all domains were low risk, the overall bias was classified as " low risk ". When at least one domain was some concerns but no high risk, the overall bias was classified as " some concerns ". Once at least one domain was high risk, the overall bias was classified as " high risk ".

## 2.7  Statistical analysis

A frequentist NMA was conducted in this study by use of Stata (version 15.0, Stata Corporation) [25]. Standard mean difference (SMD) and 95% CI were set as effect values for continuous data. And the methodological sequence was executed in accordance with the established procedural protocol outlined below: (1) a network plot was created to explore the comparative relationships between treatment regimens; (2) the design-by-treatment interaction model was performed to evaluate global inconsistency [26]; (3) the node-splitting method was conducted to test local inconsistency. When a closed loop was formed in 2-arm studies, the loop inconsistency test was performed to evaluate the consistency between direct and indirect comparisons [27]; (4) pairwise comparisons containing the SMD and 95% CI were compiled into a league table with both direct and indirect comparisons; (5) the interventions were ranked based on the SUCRA. The range of SUCRA values was from 0% to 100%, and higher SUCRA values represented more effective treatment [28]. Additionally, funnel plots were drawn for publication bias tests.

## 3  Results

### 3.1  Search results

A total of 2392 studies were identified and imported into Endnote X9 software, and 1114 duplicate articles were removed. Following the screening of titles and abstracts, 1242 irrelevant articles were excluded. After a full-text screening, 19 articles were further excluded. Finally, 17 RCTs were included [29–45]. The flow chart of the study selection process is shown in Fig 1.

### 3.2  Study characteristics and quality

This NMA included 17 RCTs from 11 countries [29–45], there were a total of 1013 participants, including fourteen 2-arm studies [29,30,32–43], two 3-arm studies [44,45], and one 4-arm study [31]. These articles were published between 2007 and 2024, but one study did not report the age of participants [32], 3 studies did not explicitly mention the scar location of participants [33,35,36], and only 10 studies reported days post-burn until enrollment [29–32,35,38–41,43]. The selected RCTs reported seventeen non-surgical treatments for post-burn scars, including MT, PT, ESWT, silicone gel (SG), $CO_2$LT, pressure therapy and silicone (PTS), orange polarized light therapy (OPLT), high intensity laser therapy (HILT), pulsed dye laser (PDL), routine treatment (RT) or their combination. In addition, 11 studies reported the outcome of VSS score [29,30,32,34,36,37,39,40,42,43,45], 7 studies reported the outcome of scar thickness [31,32,36,37,41,43,45], and 5 studies reported the outcome of VAS score [31,33,35,38,43]. Table 1 shows the characteristics of the included studies.

For overall risk of bias, 6 studies (35.3%) were judged to be at low risk of bias, 7 studies (41.2%) were judged to be at some risk of bias, 4 studies (23.5%) were judged to be at high risk of bias. The main reasons of high risk of bias were from randomization process, lacking of blinding of participants and personnel. The summary of the risk of bias assessment is shown in Fig 2.

### 3.3  Network meta-analysis

**3.3.1  VSS score.** Fig 3 shows the network plot for VSS score. This NMA incorporated data from 11 clinical studies encompassing a total of 505 participants [29,30,32,34,36,37,39,40,42,43,45], which included 6 monotherapies and 4

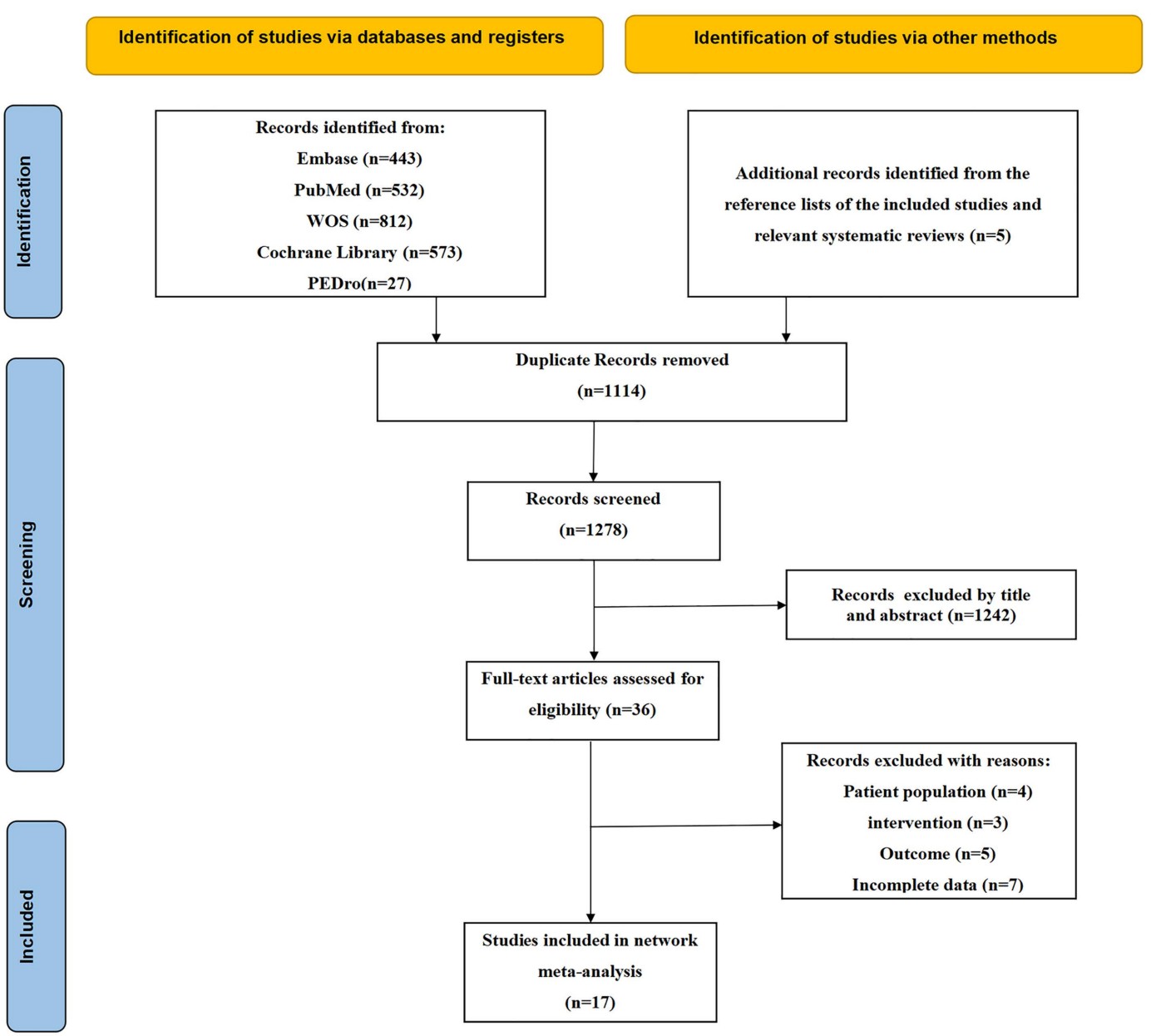

**Fig 1. Flow chart of the literature search.**

combination treatments: MT, RT, PTS, PT, $CO_2$LT, ESWT+RT, OPLT+RT, $CO_2$LT+PDL, PDL and microneedling. In the network plot, the closed loop was formed by a 3-arm study [45] therefore the consistency model was applicable for the NMA. The results of NMA for VSS score revealed that, compared with RT, MT (SMD = −1.92, 95% CI −3.30 to −0.55) and ESWT+RT (SMD = −1.01, 95% CI −1.92 to −0.10) resulted in a statistically significant reduction in VSS score ($P < 0.05$) (Table 2).

**Table 1. Characteristics of the included trials (n = 17).**

| Study | Year | country | Participants (Exp/Con) | Mean age or range (year) | Duration from injury (day) | Scar location | Intervention (Exp/con) | Outcomes |
|---|---|---|---|---|---|---|---|---|
| Roh et al [29] | 2007 | Korea | 18/17 | Exp = 33.3 ± 8.3 Con = 39.1 ± 8.2 | Exp = 127.6 ± 171.1 Con = 95.3 ± 83.7 | Forearm Hand | MT/ RT | VSS score |
| Harte et al [30] | 2009 | England | 10/9 | Exp = 38.8 ± 11.1 Con = 34.7 ± 17.8 | Exp = 98.1 ± 33.5 Con = 125.1 ± 40.6 | Upper limb Lower limb | PTS/ PT | VSS score |
| Li-Tsang et al [31] | 2010 | China | 24/26/22/12 | 21.8 ± 18.7 | 447 ± 924 | Limbs Body | PTS + RT/ PT + RT/ SG + RT/ RT | VAS score Scar thickness |
| Li-Tsang et al [32] | 2010 | China | 36/9 | not reported | 90.0 | Limb | PT/ RT | VSS score Scar thickness |
| Parlak et al [33] | 2010 | Turkey | 32/31 | 14.07 ± 1.78 | not reported | not reported | MT + RT/ RT | VAS score |
| Steinstraesser et al [34] | 2011 | German | 19/19 | >18 | not reported | Anterior trunk Upper limb Lower limb | PTS/ PT | VSS score |
| Cho et al [35] | 2014 | Korea | 76/70 | Exp = 46.06 ± 8.63 Con = 47.21 ± 8.22 | Exp = 148.77 ± 56.85 Con = 156.47 ± 56.48 | not reported | MT + RT/ RT | VAS score |
| Blome-Eberwein et al [36] | 2016 | USA | 36 | 39.00 ± 15.65 | not reported | not reported | $CO_2$LT/ RT | VSS score Scar thickness |
| Zaghloul et al [37] | 2016 | Egypt | 20/20 | 20-45 | not reported | Body | ESWT + RT/ RT | VSS score Scar thickness |
| Ebid et al [38] | 2017 | Saudi Arabia | 24/25 | Exp = 30.25 ± 12.05 Con = 32.45 ± 11.21 | Exp = 33.46 ± 3.38 Con = 34.67 ± 2.45 | Upper limbs Abdomen | HILT/ RT | VAS score |
| Elrashid et al [39] | 2018 | Egypt | 15/15 | Exp = 4.66 ± 1.63 Con = 5.26 ± 1.98 | >60.0 | Wrist, Hands | OPLT + RT/ RT | VSS score |
| Ouyang et al [40] | 2018 | China | 28/28 | 3-51 | <90.0 | Face, Neck, Body, Arms, Legs | $CO_2$LT + PDL/ PDL | VSS score |
| Nedelec et al [41] | 2019 | Canada | 60 | >18 | 34 - 768 | Upper limb, Hand, Torso, Lower limb, Foot | MT + RT/ RT | Scar thickness |
| Golnaz et al [42] | 2019 | Iran | 30/30 | Exp = 32 ± 5 Con = 37 ± 4 | not reported | Arm, Forearm, Trunk | Microneedling/ $CO_2$LT | VSS score |
| Joo et al [43] | 2020 | Korea | 23/25 | Exp = 47.09 ± 11.09 Con = 48.56 ± 11.18 | <180.0 | Right hand | ESWT+ RT/ RT | VAS score VSS score Scar thickness |
| Wiseman et al [44] | 2020 | Australia | 48/51/43 | 0-18 | not reported | Torso, Upper limb, Lower limb | PTS/ SG/ PT | Scar thickness |
| Kivi et al [45] | 2024 | Iran | 25/19/18 | Exp = 23.87 ± 4.11 Con = 21.84 ± 4.23 Con = 24.11 ± 4.66 | not reported | Face, Neck, Chest, Abdomen, Upper limb, Lower limb | $CO_2$LT + PDL/ $CO_2$LT/ PDL | VSS score |

Exp: Experiment, Con: Control, Massage Therapy: MT, Routine Treatment: RT, Pressure Garments and Silicone: PTS, Pressure Therapy: PT, Silicone Gel: SG, $CO_2$ Laser Therapy: $CO_2$ LT, Extracorporeal Shock Wave Therapy: ESWT, High Intensity Laser Therapy: HILT, Pulsed dye laser: PDL, Orange Polarized Light Therapy: OPLT.

The SUCRA values of the 10 treatments were as follows: MT (89.0%)> ESWT+RT (63.5%)> OPLT+RT (63.0%)> microneedling (58.5%)> PT (57.9%)> PTS (57.3%)> $CO_2$LT+PDL (39.9%)> $CO_2$LT (28.6%)> PDL (23.5%)> RT (18.8%) (Fig 4).

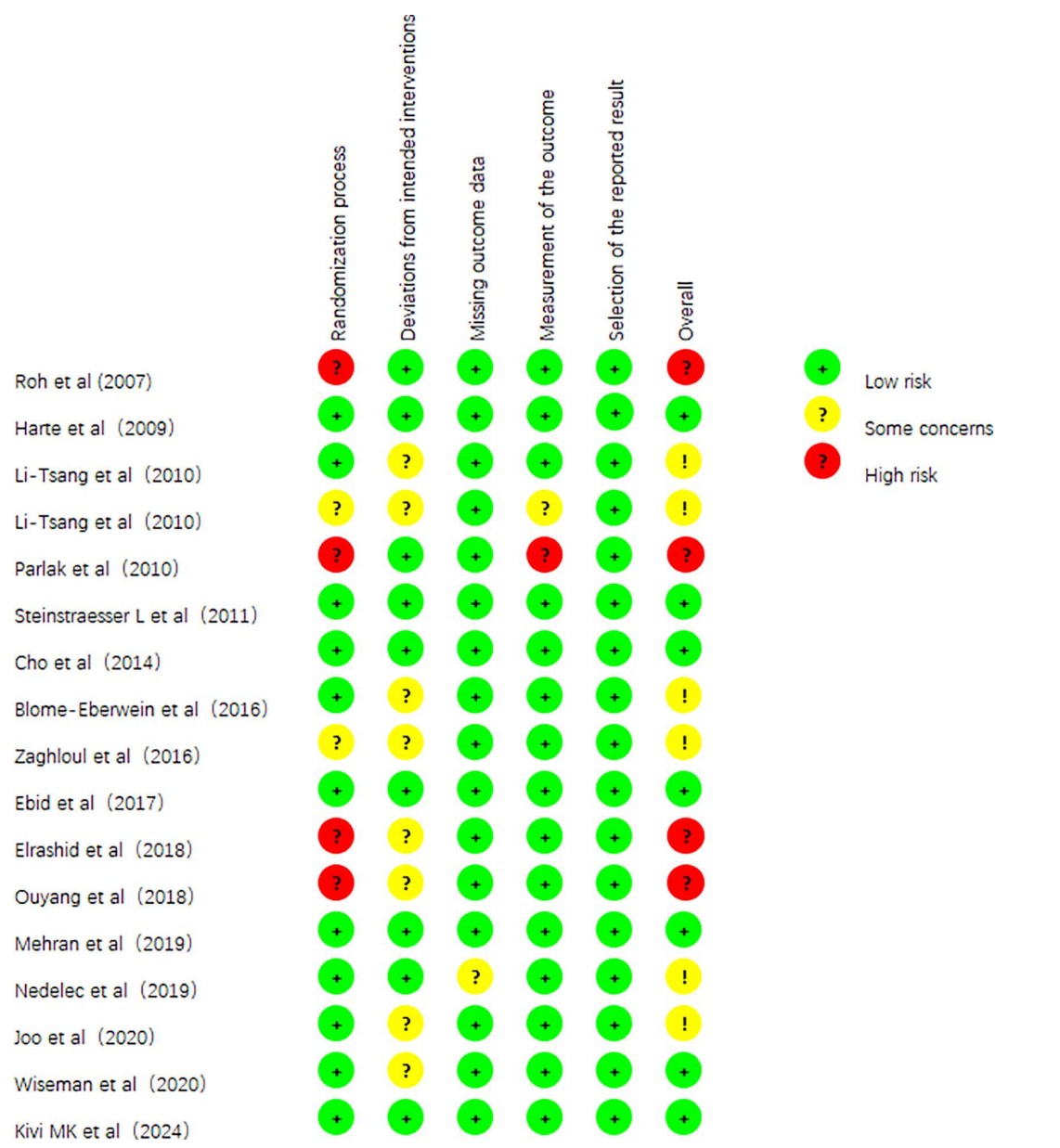

**Fig 2. Risk of bias graph.**

In summary, the results of NMA and SUCRA rankings indicated that MT was the most effective non-surgical treatment in reducing VSS score, followed by ESWT＋RT. In addition, the funnel plot showed symmetry indicating no publication bias (Fig 5).

**3.3.2 Scar thickness.** Fig 6 presents the network plot for scar thickness. A total of 7 studies reported scar thickness outcomes [31,32,33,37,41,43,44], including 4 monotherapies and 3 combination treatments: PTS, PT, SG, RT, $CO_2$LT, ESWT+RT and MT＋RT. In the network plot, the closed loops was formed by a 4-arm study and a 3-arm study [31,44], therefore the consistency model was applicable for the NMA. The results of NMA for scar thickness indicated that the difference was statistically significant ($P<0.05$) between $CO_2$LT and RT (SMD＝−1.75, 95% CI-2.65 to −0.84), ESWT+RT

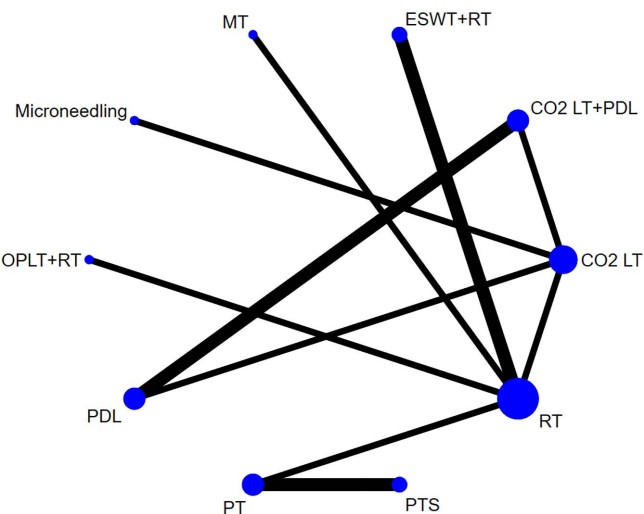

**Fig 3. Network plot for VSS score.**

**Table 2. Results of network meta-analysis for vancouver scar scale score based on the frequentist approach (SMD, 95% CI).**

| MT | | | | | | | | | |
|---|---|---|---|---|---|---|---|---|---|
| −0.91 (−2.56,0.74) | **ESWT+RT** | | | | | | | | |
| −0.89 (−2.82,1.04) | 0.02 (−1.60,1.65) | **OPLT+RT** | | | | | | | |
| −1.06 (−3.26,1.14) | −0.15 (−2.09,1.79) | −0.17 (−2.35,2.01) | **Micronee-dling** | | | | | | |
| −1.03 (−2.95,0.89) | −0.12 (−1.74,1.50) | −0.15 (−2.05,1.75) | 0.03 (−2.15,2.20) | **PT** | | | | | |
| −1.03 (−3.18,1.11) | −0.12 (−2.00,1.76) | −0.14 (−2.27,1.98) | 0.03 (−2.34,2.40) | 0.00 (−0.95,0.96) | **PTS** | | | | |
| −1.55 (−3.72,0.62) | −0.64 (−2.55,1.27) | −0.66 (−2.82,1.49) | −0.49 (−2.19,1.21) | −0.52 (−2.67,1.63) | −0.52 (−2.87,1.83) | **CO$_2$LT+PDL** | | | |
| −1.73 (−3.56,0.09) | −0.82 (−2.32,0.69) | −0.84 (−2.65,0.96) | −0.67 (−1.89,0.55) | −0.70 (−2.49,1.10) | −0.70 (−2.73,1.33) | −0.18 (−1.36,1.00) | **CO$_2$LT** | | |
| −1.90 (−4.08,0.28) | −0.99 (−2.90,0.93) | −1.01 (−3.17,1.15) | −0.84 (−2.55,0.87) | −0.86 (−3.02,1.29) | −0.87 (−3.22,1.49) | −0.35 (−1.23,0.54) | −0.17 (−1.36,1.03) | **PDL** | |
| **−1.92 (−3.30,-0.55)** | **−1.01 (−1.92,-0.10)** | −1.03 (−2.38,0.31) | −0.86 (−2.58,0.85) | −0.89 (−2.23,0.45) | −0.89 (−2.54,0.75) | −0.37 (−2.05,1.31) | −0.19 (−1.39,1.01) | −0.03 (−1.72,1.66) | **RT** |

MT: Massage Therapy, ESWT+RT: Extracorporeal Shock Wave Therapy+Routine Treatment, OPLT+RT: Orange Polarized Light Therapy+Routine Treatment, PT: Pressure Therapy, PTS: Pressure Garments and Silicone, CO$_2$ LT+PDL: CO$_2$ Laser Therapy+Pulsed dye laser, CO$_2$LT: CO$_2$ Laser Therapy, PDL: Pulsed dye laser, RT: Routine Treatment. The off-diagonal cells in the league table display the relative treatment effects (standard mean difference and 95% confidence intervals) for pairwise comparisons estimated in the network meta-analysis.

and RT (SMD=−1.14, 95% CI-1.82 to −0.45), CO$_2$LT and MT+RT (SMD=−1.68, 95% CI-2.90 to −0.47), ESWT+RT and MT+RT (SMD=−1.07, 95% CI-2.13 to −0.01), CO$_2$LT and PTS (SMD=−1.54, 95% CI −2.84 to −0.25), CO$_2$LT and PT (SMD=−1.52, 95% CI −2.81 to −0.23), and CO$_2$LT and SG (SMD=−1.41, 95% CI −2.70 to −0.11) (Table 3).

The SUCRA values suggested that the 7 treatments were ranked as follows: CO$_2$LT (96.8%)> ESWT+RT (82.0%)> SG (47.5%)> PT (37.6%)> PTS (35.4%)> MT+RT (28.6%)> RT (22.1%) (Fig 7).

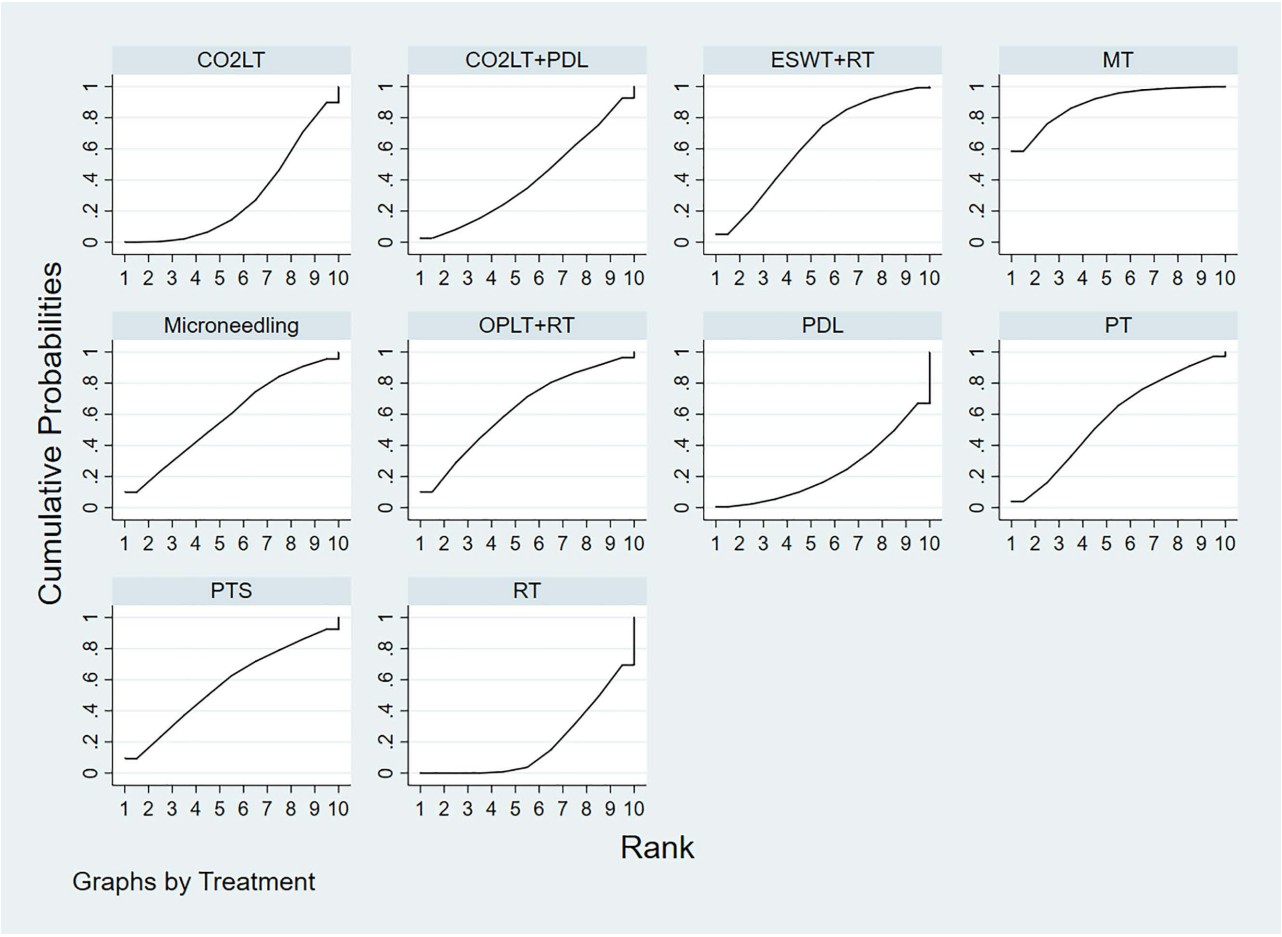

**Fig 4. Plot of the surface under the cumulative ranking curve for VSS score.** $CO_2$LT: $CO_2$ Laser Therapy (the SUCRA value is 28.6%), $CO_2$LT+PDL: $CO_2$ Laser Therapy+Pulsed dye laser, ESWT+RT (the SUCRA value is 63.5%): Extracorporeal Shock Wave Therapy+Routine Treatment (the SUCRA value is 63.5%), MT: Massage Therapy (the SUCRA value is 89.0%), OPLT+RT: Orange Polarized Light Therapy+Routine Treatment (the SUCRA value is 63.0%), PDL: Pulsed dye laser (the SUCRA value is 23.5%), PT: Pressure Therapy (the SUCRA value is 57.9%), PTS: Pressure Garments and Silicone (the SUCRA value is 57.3%), RT: Routine Treatment (the SUCRA value is 18.8%). The overall position of the curve corresponds with the SUCRA value reported in the results. A higher curve indicates better performance in terms of treatment ranking.

In summary, the results of NMA and SUCRA rankings indicated that $CO_2$LT was the most effective non-surgical treatment in reducing scar thickness, followed by ESWT+RT. And the funnel plot indicated that there was no publication bias (Fig 8).

**3.3.3 VAS score.** Fig 9 showed the network plot for VAS score. In the NMA, 5 studies [31,33,35,38,43] with 390 patients were evaluated, including 2 monotherapies and 5 combination treatments: PTS+RT, PT+RT, SG+RT, RT, MT+RT, HILT and ESWT+RT. In the network plot, the closed loop was formed by a 4-arm study [31], and the consistency model was applicable for the NMA. The results of NMA for VAS score indicated that there was no statistical significance in the pairwise comparison among the 7 non-surgical treatments ($P > 0.05$) (Table 4).

The SUCRA values suggested that the 7 treatments were ranked as follows: ESWT+RT (58.6%)> HILT (57.6%)> PTS+RT (56.1%)> SG+RT (52.0%)> RT (47.9%)> PT+RT (46.2%)> MT+RT (31.6%) (Fig 10).

In summary, the results of NMA and SUCRA rankings suggested that ESWT+RT was the most effective non-surgical treatment in reducing VAS score. And the funnel plot showed no indication of the presence of publication bias. (Fig 11).

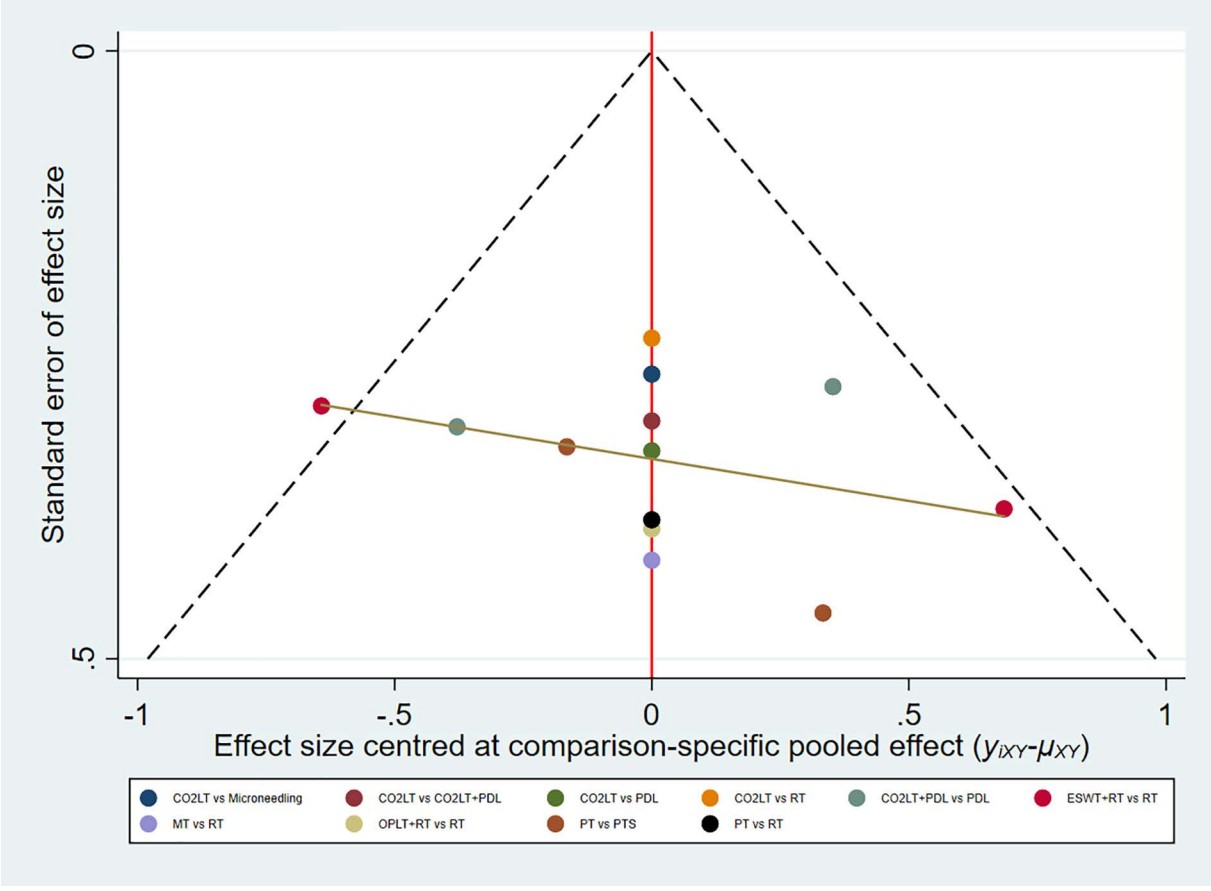

**Fig 5. Funnel plot for VSS score.**

## 4 Discussions

This NMA provides the first comprehensive comparison of non-surgical therapies targeting outcomes related to post-burn scars. Our findings reveal MT emerged as the most effective intervention for improving VSS score, $CO_2LT$ for reducing scar thickness, and ESWT+RT for alleviating pain. These results not only validate current clinical practices but also offer mechanistic insights into how different modalities address specific pathophysiological aspects of post-burn scarring.

According to our findings, MT was identified as the most effective non-surgical intervention for reducing VSS) score. This conclusion is supported by the network meta-analysis, which demonstrated that MT significantly outperformed other treatments in improving overall scar quality. Specifically, MT was associated with a marked improvement in scar pliability, a critical component of the VSS score, while also showing potential benefits in addressing other aspects such as scar vascularity, pigmentation, and thickness [21,22]. The mechanisms underlying the efficacy of MT may include its ability to promote collagen remodeling, enhance tissue elasticity, and reduce scar stiffness through mechanical manipulation [41]. These effects contribute to the overall improvement in scar appearance and functionality.

The included studies reported that MT is typically performed by trained massage therapists to treat post-burn scars, with sessions conducted 1–3 times per week over a period of 4–12 weeks [29,33,35,41]. However, variability in the qualifications of massage therapists and the lack of standardized protocols across studies were noted, which may limit the

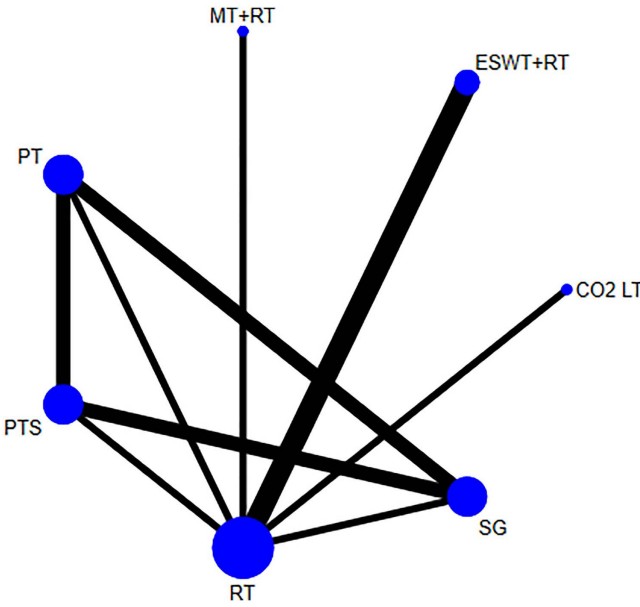

**Fig 6. Network plot for scar thickness.**

**Table 3. Results of network meta-analysis for scar thickness based on the frequentist approach (SMD, 95% CI).**

| $CO_2LT$ | | | | | | |
|---|---|---|---|---|---|---|
| −0.61 (−1.75,0.53) | **ESWT+RT** | | | | | |
| **−1.41 (−2.70,-0.11)** | −0.80 (−1.95,0.36) | **SG** | | | | |
| **−1.52 (−2.81,-0.23)** | −0.91 (−2.06,0.24) | −0.11 (−0.73,0.50) | **PT** | | | |
| **−1.54 (−2.84,-0.25)** | −0.93 (−2.08,0.22) | −0.14 (−0.76,0.48) | −0.03 (−0.65,0.59) | PTS | | |
| **−1.68 (−2.90,-0.47)** | **−1.07 (−2.13,-0.01)** | −0.28 (−1.51,0.95) | −0.17 (−1.39,1.06) | −0.14 (−1.37,1.09) | **MT+RT** | |
| **−1.75 (−2.65,-0.84)** | **−1.14 (−1.82,-0.45)** | −0.34 (−1.26,0.58) | −0.23 (−1.15,0.69) | −0.20 (−1.13,0.72) | −0.06 (−0.87,0.75) | **RT** |

$CO_2LT$: $CO_2$ Laser Therapy, ESWT+RT: Extracorporeal Shock Wave Therapy+Routine Treatment, SG: Silicone Gel, PT: Pressure Therapy, PTS: Pressure Garments and Silicone, MT+RT: Massage Therapy+Routine Treatment, RT: Routine Treatment. The off-diagonal cells in the league table display the relative treatment effects (standard mean difference and 95% confidence intervals) for pairwise comparisons estimated in the network meta-analysis.

reproducibility and consistency of MT outcomes in clinical practice. This highlights the need for more rigorous standardization of MT techniques and training requirements to ensure optimal therapeutic results.

These findings underscore the value of MT as a first-line non-surgical therapy for managing post-burn scars. Nevertheless, further research is warranted to refine application protocols, establish standardized guidelines, and evaluate the efficacy of MT across diverse patient populations and scar types. Such efforts will help maximize the therapeutic potential of MT and improve clinical decision-making for burn scar management.

The NMA demonstrated that ESWT+RT is a promising treatment for post-burn scars, particularly in reducing scar thickness and VSS score. Additionally, the NMA indicated that ESWT+RT was the most effective intervention for improving VAS score, highlighting its potential in alleviating pain associated with burn scars. ESWT induces microtrauma in scar tissue, breaks down collagen fibers, and releases growth and molecular factors, which contribute to scar remodeling and tissue repair [23]. Additionally, ESWT has positive effects on pain receptor physiology, leading to pain relief [46]. These mechanisms collectively improve the overall appearance and functionality of post-burn scars.

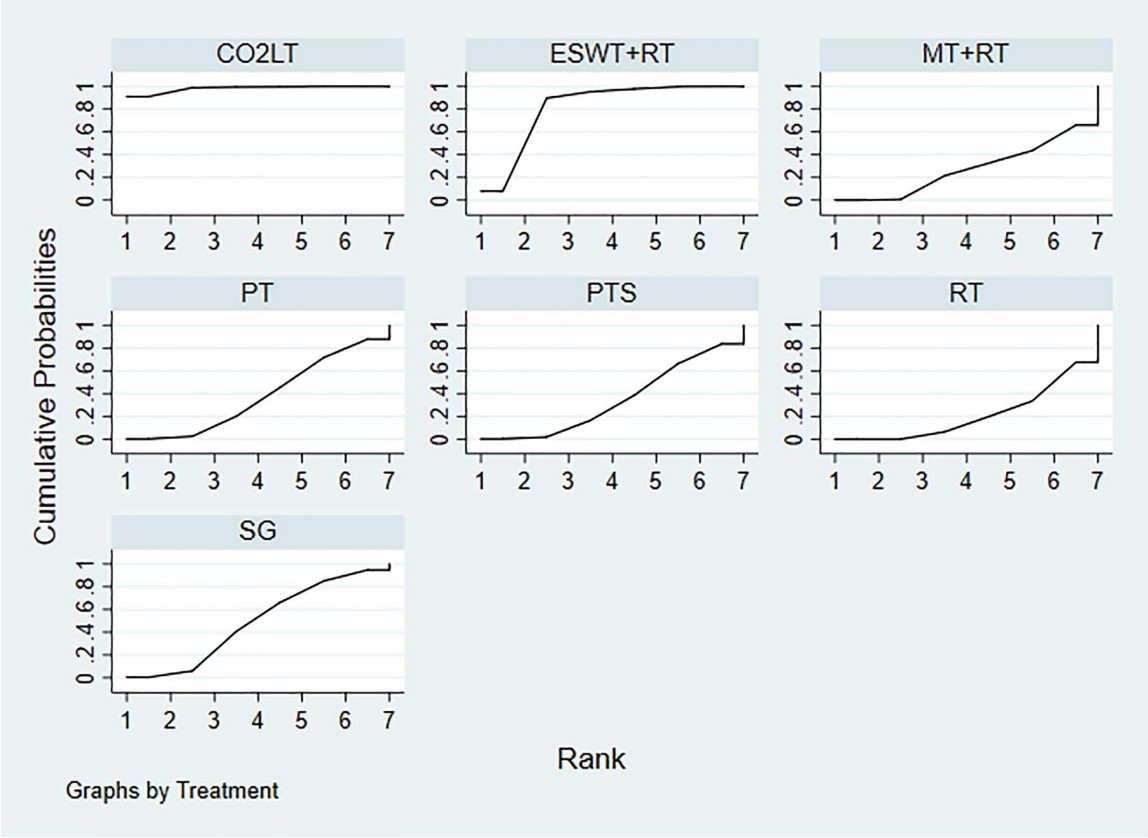

**Fig 7. Plot of the surface under the cumulative ranking curve for scar thickness.** CO₂LT: CO₂ Laser Therapy (the SUCRA value is 96.8%), ESWT+RT: Extracorporeal Shock Wave Therapy+Routine Treatment (the SUCRA value is 82.0%), MT+RT: Massage Therapy+Routine Treatment (the SUCRA value is 28.6%), PT: Pressure Therapy (the SUCRA value is 37.6%), PTS: Pressure Garments and Silicone (the SUCRA value is 35.4%), SG: Silicone Gel (the SUCRA value is 47.5%), RT: Routine Treatment (the SUCRA value is 22.1%). The overall position of the curve corresponds with the SUCRA value reported in the results. A higher curve indicates better performance in terms of treatment ranking.

In the included studies, ESWT was typically administered at an intensity of 100 impulses/cm², with each session delivering 1000–3000 impulses over 4–6 sessions conducted at one-week intervals [37,43]. Zaghloul et al. and Joo et al. reported that ESWT+RT had a superior effect on improving VSS score compared to other treatments [37,43]. Similarly, Aguilera-Sáez et al. noted that using ESWT could enhance the appearance and alleviate pain of burn scars, which aligns with our findings [23]. Furthermore, a meta-analysis of RCTs [47] revealed that combining EWST with comprehensive rehabilitation therapy had a better therapeutic effect on post-burn pathological scars than comprehensive rehabilitation therapy alone, without obvious side effects. This evidence supports the integration of ESWT+RT into clinical practice as a safe and effective treatment option.

In conclusion, ESWT+RT represents a first-line therapeutic option for managing post-burn scars due to its demonstrated efficacy in reducing scar thickness, improving VSS scores, and alleviating pain. However, further research is needed to optimize treatment protocols, standardize application methods, and evaluate long-term outcomes across diverse patient populations.

Our NMA revealed that CO₂LT is the most effective non-surgical treatment for reducing scar thickness. This finding is consistent with previous studies that have demonstrated CO₂LT's ability to promote collagen remodeling and improve skin texture [36,48]. By delivering high-energy laser pulses to the affected area, CO₂LT induces controlled thermal damage to

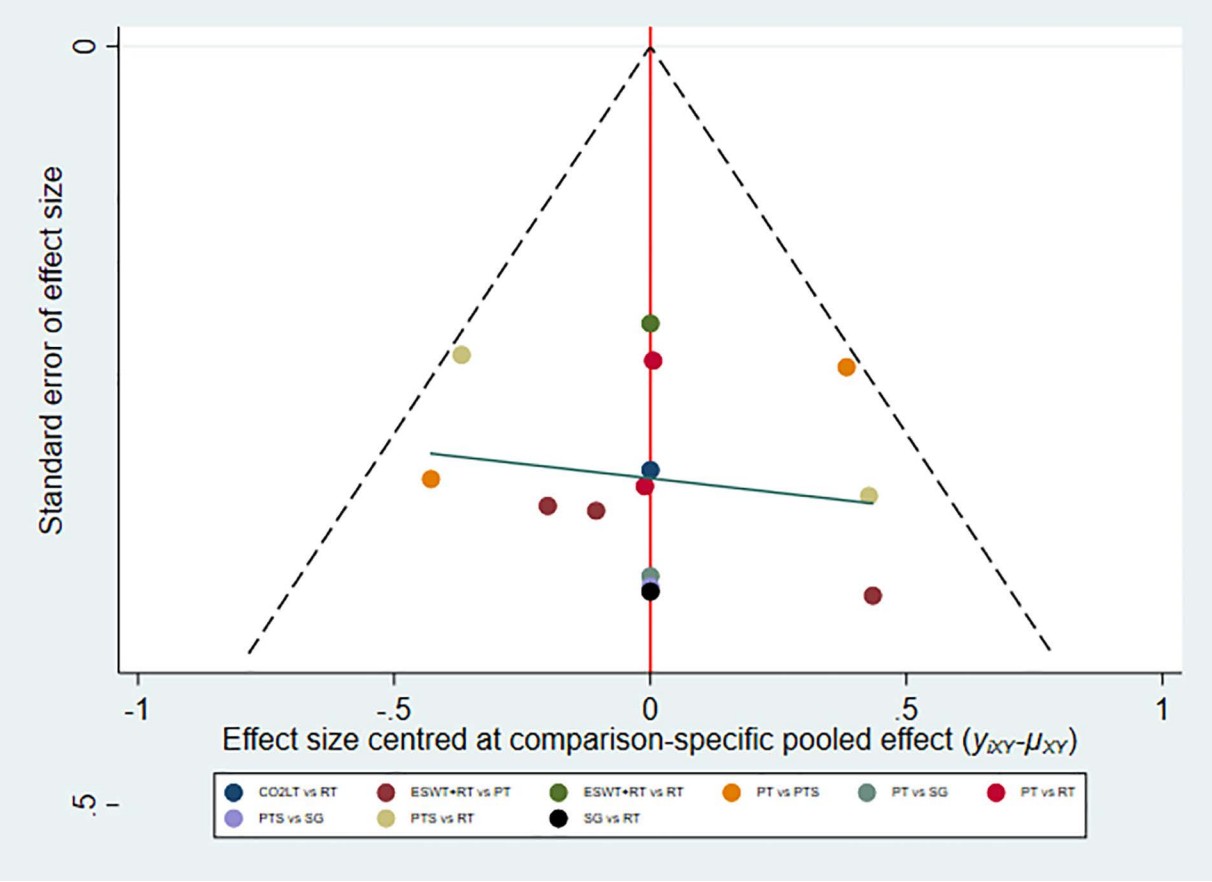

**Fig 8. Funnel plot for scar thickness.**

the dermis, stimulating the production of new collagen and elastin fibers [49]. This process not only reduces scar thickness but also improves the overall appearance and functionality of the scar tissue. The significant reduction in scar thickness observed in our NMA underscores the therapeutic potential of $CO_2LT$ as a first-line treatment for post-burn scars.

The effectiveness of $CO_2LT$ can be attributed to its unique mechanism of action, which involves both ablative and non-ablative processes [50]. In ablative $CO_2LT$, the laser removes thin layers of the epidermis and dermis, leading to immediate tissue contraction and subsequent collagen remodeling. Non-ablative $CO_2LT$, on the other hand, targets deeper layers of the dermis without damaging the epidermis, promoting gradual collagen regeneration and scar improvement. Both approaches contribute to the reduction in scar thickness by breaking down excessive collagen deposits and promoting the formation of more organized and flexible scar tissue [48]. Additionally, the heat generated by the laser stimulates the release of growth factors and cytokines, further enhancing the healing process. These mechanisms collectively explain why $CO_2LT$ outperforms other non-surgical treatments in reducing scar thickness and improving scar quality.

Given the superior efficacy of $CO_2LT$ in reducing scar thickness, it should be considered a primary option for managing post-burn scars. However, it is important to recognize that $CO_2LT$ requires specialized equipment and trained professionals to administer the treatment safely and effectively. Therefore, standardization of treatment protocols and training guidelines is essential to ensure optimal outcomes across different clinical settings. Moreover, while our findings highlight the benefits of $CO_2LT$, further research is needed to explore its long-term effects and assess its efficacy in diverse patient

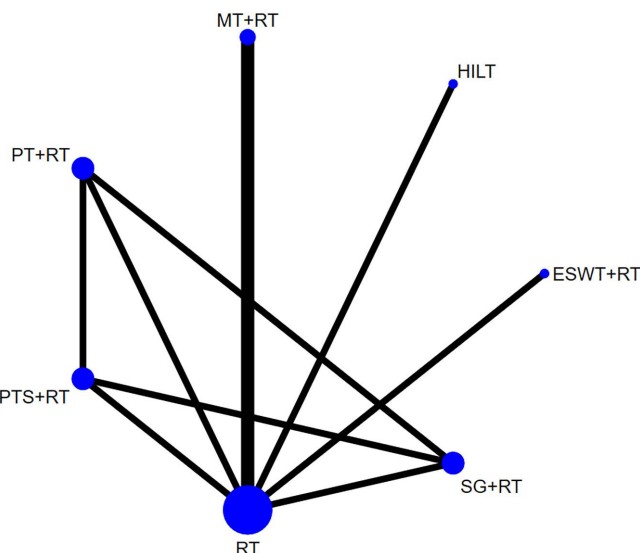

**Fig 9. Network plot for VAS score.**

**Table 4. Results of network meta-analysis for visual analog scale score based on the frequentist approach (SMD, 95% CI).**

| | | | | | | |
|---|---|---|---|---|---|---|
| **ESWT + RT** | | | | | | |
| −0.08 (−8.90,8.73) | **HILT** | | | | | |
| −0.34 (−9.17,8.48) | −0.26 (−9.19,8.67) | **PTS+RT** | | | | |
| −0.61 (−9.44,8.21) | −0.53 (−9.46,8.40) | −0.27 (−6.59,6.04) | **SG+RT** | | | |
| −0.99 (−7.18,5.20) | −0.91 (−7.22,5.40) | −0.65 (−6.97,5.67) | −0.38 (−6.70,5.94) | **RT** | | |
| −1.11 (−9.94,7.71) | −1.03 (−9.96,7.90) | −0.77 (−7.09,5.54) | −0.50 (−6.81,5.81) | −0.12 (−6.44,6.20) | **PT+RT** | |
| −2.28 (−9.89,5.33) | −2.19 (−9.92,5.53) | −1.94 (−9.67,5.80) | −1.66 (−9.40,6.07) | −1.29 (−5.75,3.18) | −1.16 (−8.90,6.57) | **MT+RT** |

ESWT + RT: Extracorporeal Shock Wave Therapy+Routine Treatment, HILT: High Intensity Laser Therapy, PTS+RT: Pressure Garments and Silicone+Routine Treatment, SG+RT: Silicone Gel+Routine Treatment, RT: Routine Treatment, PT+RT: Pressure Therapy+Routine Treatment, MT+RT: Massage Therapy+Routine Treatment. Area under the curve is positively correlated with the best ordering. The off-diagonal cells in the league table display the relative treatment effects (standard mean difference and 95% confidence intervals) for pairwise comparisons estimated in the network meta-analysis.

populations. In addition, comparative studies evaluating the cost-effectiveness of $CO_2LT$ relative to other treatments could provide valuable insights for healthcare providers and policymakers. By addressing these areas, future studies can help refine clinical guidelines and enhance the management of post-burn scars.

In this NMA, we integrated all available evidence from 17 RCTs involving 17 different non-surgical treatments. The analyses revealed no evidence of publication bias, and all included articles demonstrated a low risk of methodological bias. The results could offer valuable guidance for medical staff in devising treatment strategies.

Although our analysis focused on commonly reported quantitative outcomes such as VSS and VAS scores, we acknowledge certain limitations in these scales with regard to capturing the full complexity of scar characteristics and the patient's subjective experience. The Patient and Observer Scar Assessment Scale (POSAS), which incorporates both clinician-observed and patient-reported assessments across multiple domains—including color, thickness, pliability, and symptom perception—offers a more comprehensive and holistic alternative. To enhance the validity and clinical relevance

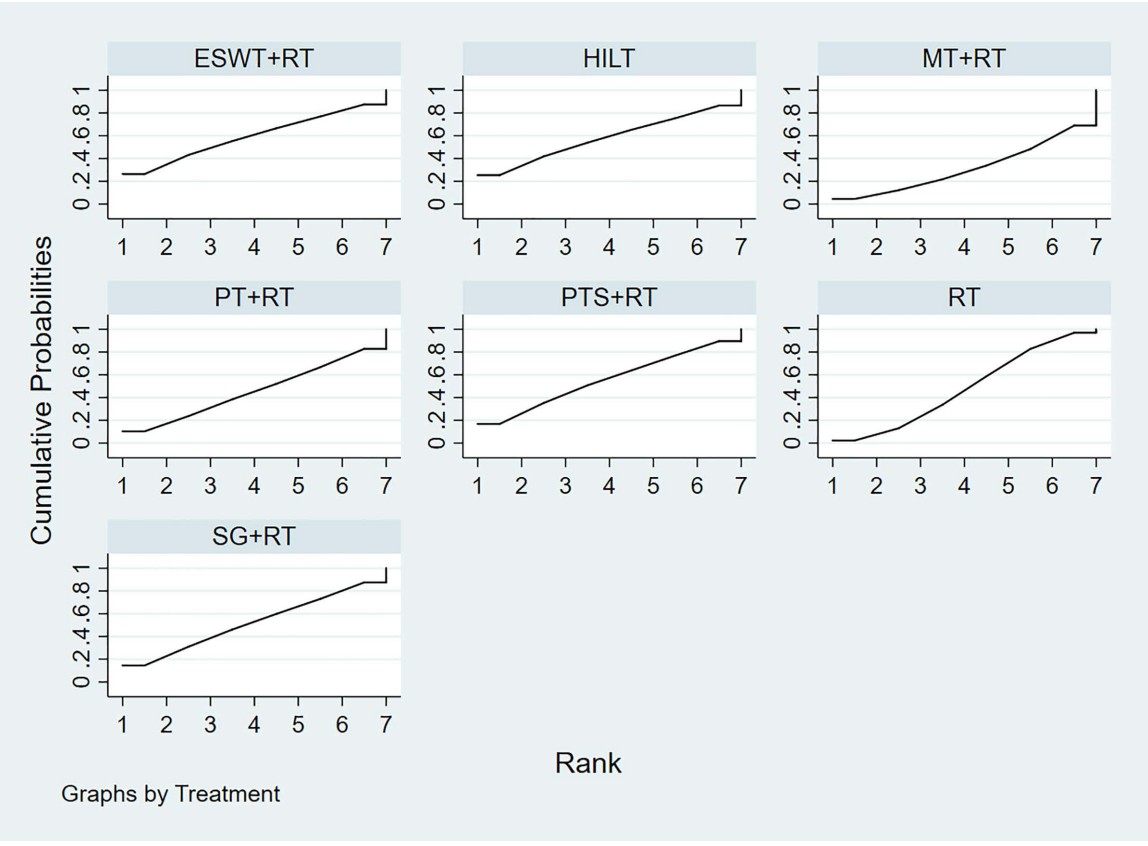

**Fig 10. Plot of the surface under the cumulative ranking curve for VAS score.** ESWT+RT: Extracorporeal Shock Wave Therapy+Routine Treatment (the SUCRA value is 58.6%), HILT: High Intensity Laser Therapy (the SUCRA value is 57.6%), MT+RT: Massage Therapy+Routine Treatment (the SUCRA value is 31.6%), PT+RT: Pressure Therapy+Routine Treatment (the SUCRA value is 46.2%), PTS+RT: Pressure Garments and Silicone+Routine Treatment (the SUCRA value is 56.1%), SG+RT: Silicone Gel+Routine Treatment (the SUCRA value is 52.0%), RT: Routine Treatment (the SUCRA value is 47.9%). The overall position of the curve corresponds with the SUCRA value reported in the results. A higher curve indicates better performance in terms of treatment ranking.

of future research, we recommend that systematic reviews and clinical studies prioritize the use of standardized, multidimensional outcome tools like POSAS in the evaluation of post-burn scar interventions.

Additionally, in line with the Global Top 10 Research Priorities for Burn Care [51], this NMA identified several critical gaps in the current evidence base: (1) a lack of long-term follow-up data on the efficacy and safety of non-surgical treatments; (2) underutilization of patient-reported outcome measures (PROMs), such as POSAS, which limits understanding of patient-centered impacts beyond clinician-reported metrics like VSS or VAS; (3) limited reporting of cost-effectiveness or resource implications, which hinders informed decision-making in low- and middle-income settings; (4) insufficient standardization of outcome reporting across trials, contributing to heterogeneity and limiting comparability.

Future research should prioritize addressing these gaps to generate more robust, equitable, and clinically meaningful evidence in post-burn scar management. Specifically, we recommend conducting economic evaluations of promising interventions—particularly in resource-limited settings—and developing core outcome sets for post-burn scar interventions to support evidence synthesis and guideline development.

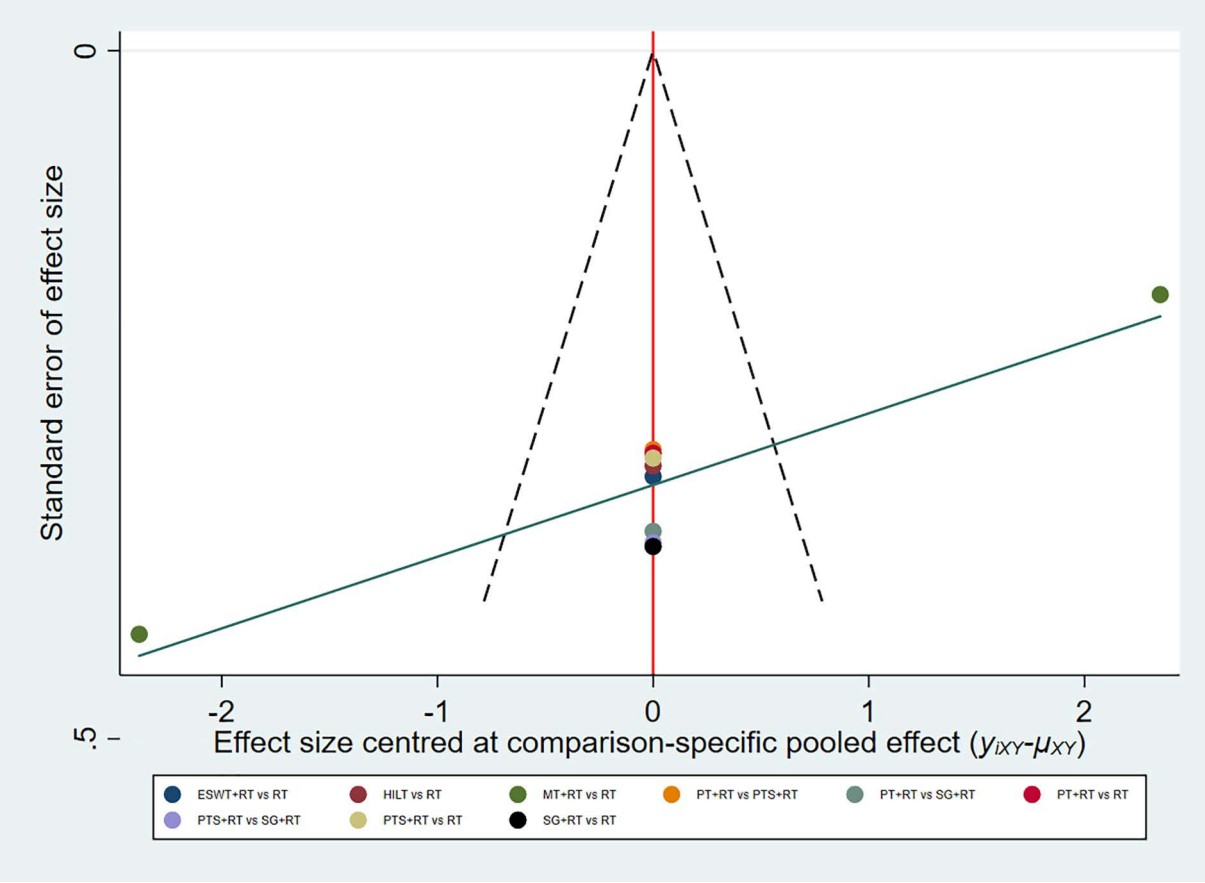

**Fig 11. Funnel plot for VAS score.**

## 5 Strengths and limitations

Some limitations of this study also must be acknowledged. Firstly, few studies were included with a lack of large sample sizes and multicenter RCTs, therefore caution should be exercised when interpreting the results. Secondly, some original studies lacked comprehensive details regarding routine treatment. Furthermore, discrepancies were observed in various non-surgical treatments with regard to the frequency of sessions and the intervals for follow-up assessments. This NMA focuses on outcomes assessed at approximately 6 months post-injury. This timepoint may not reflect the optimal timing for all interventions (such as $CO_2LT$), and it may limit the number of eligible studies reporting outcomes within this window. As a result, the estimated effect sizes may not fully represent the potential benefits of interventions delivered earlier or later in the recovery trajectory. In addition, one limitation of this NMA is the exclusion of studies reporting only POSAS outcomes, which may have led to the omission of more recent, contemporary interventions that adopt this more comprehensive and patient-centered assessment tool. This highlights the need for future studies to harmonize outcome reporting and adopt standardized, multidimensional scales such as POSAS to enhance the comparability and relevance of evidence in post-burn scar management. Finally, the data on duration from injury in Table 1 was incomplete with some differences in detailed data which might be potential sources of heterogeneity.

It should be noted that all interventions not involving excisional surgery were classified as "non-surgical" in this review. This definition intentionally includes ablative $CO_2$ laser and fully ablative microneedling procedures, even when performed

under general anaesthesia. Though these modalities differ from other non-surgical interventions in terms of injury depth, perioperative requirements, and expected morbidity, a consistent classification was adopted to enable a comprehensive synthesis of the available evidence on non-surgical approaches for post-burn scar management. However, this approach inevitably introduces clinical heterogeneity across the included studies, which may affect the precision and generalizability of the pooled estimates. Readers are therefore advised to interpret the findings with due consideration of the inherent differences among the non-surgical interventions included.

## 6 Conclusions

The result of this NMA illustrates that MT, $CO_2$LT and ESWT+RT are the most effective non-surgical treatments for reducing VSS score, scar thickness and VAS score, respectively. However, it is important to note that this conclusion is drawn from a limited number of studies. And the findings reflect outcomes at a specific stage of scar maturation (approximately 6 months post-burn). It is advocated for future trials to adopt validated multidimensional tools like POSAS to enhance consistency and reliability in scar evaluation. Additionally, well-designed randomized controlled trials with a large sample size are needed to validate these findings in the future.

## Supporting information

**S1 Table. PRISMA 2020 checklist.**
(DOCX)

**S2 File. Search strategy.**
(DOCX)

**S3 File. Extracted and analyzed data.**
(DOCX)

## Author contributions

**Conceptualization:** Xiaojuan Yang, Xiaorong Mao.

**Data curation:** Shuangying Gong, Yu Fan.

**Methodology:** Xiaorong Mao, Chunyi Xu.

**Software:** Xiaojuan Yang, Qing Wen, Xiaotao Xu.

**Supervision:** Rong Li, Wei Li.

**Writing – original draft:** Xiaojuan Yang, Rong Li.

**Writing – review & editing:** Wei Li.

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
