## [Decision Letter · Decision Letter 0]

4 Mar 2025

PONE-D-24-50779Non-surgical treatments for post-burn scars: a network meta-analysisPLOS ONE

Dear Dr. Li,

Thank you for submitting your manuscript to PLOS ONE. After careful consideration, we feel that it has merit but does not fully meet PLOS ONE’s publication criteria as it currently stands. Therefore, we invite you to submit a revised version of the manuscript that addresses the points raised during the review process.

We look forward to receiving your revised manuscript.

Kind regards,

Steven E. Wolf, MD

Academic Editor

PLOS ONE

“This study was supported by the Sichuan Science and Technology Program, China (Grant no. 23ZDYF1907).”

4. As required by our policy on Data Availability, please ensure your manuscript or supplementary information includes the following:

Reviewers' comments:

Reviewer's Responses to Questions

**Comments to the Author**

1. Is the manuscript technically sound, and do the data support the conclusions?

Reviewer #1: Yes

Reviewer #2: Partly

2. Has the statistical analysis been performed appropriately and rigorously? 

Reviewer #1: Yes

Reviewer #2: I Don't Know

3. Have the authors made all data underlying the findings in their manuscript fully available?

Reviewer #1: Yes

Reviewer #2: Yes

4. Is the manuscript presented in an intelligible fashion and written in standard English?

Reviewer #1: Yes

Reviewer #2: Yes

5. Review Comments to the Author

Reviewer #1: Well done on your hard work in producing this paper.

The paper includes impressive statical analysis and offers new evidence that would add towards burn scar care. However, I feel this paper is written poorly and needs thorough revisions to improve its overall message and strength its evidence.

The paper feels informal in nature, and I would advise the authors to remove ‘we’ in the context of the authors decision throughout the paper. Please note not to start sentences with a number. Furthermore, I feel the authors writing style could be more concise to allow for more characters to defend and boarded their discussion points.

Please note: ‘hyperplastic scar’ is the umbrella terminology for keloid, hypertrophic and cheloid scars. When the authors refer to keloid and hyperplastic scars in the text they should discuss as just hyperplastic scarring or keloid and hypertrophic scars.

Finally, despite referencing Table 1 (characteristics) and 3 – no tables have been attached to the submission. These would have to be reviewed before final acceptance. I would also hope a table offering a brief overview of each paper would add to the understanding of the paper.

Please find some more specific points of guidance per section:

Abstract

Background and aim: ‘This study aims to compare the outcomes of non-surgical treatmentS for post-burn scars using network meta-analysis.’ This will correlate to your title.

Introduction

I feel the first paragraph should be re-written to deliver a more gripping and comprehensive overview of burn scar care.

More explanation concerning the benefits of non-surgical technique should be explored to highlight the importance of what your evidence will influence in burn care.

Many sweeping statements are made addressing the reliability and validity of VSS and VAS. These tools are widely debated as to their reliability and validity and this is the main reason centres now choose the POSAS score to objectively measure scar care. The authors need to add more evidence to support their conclusion concerning VSS and VAS.

Again, a statement has been made concerning previous studies have ‘some limitations’ with no supporting discussion. If the authors want to improve from previous literature, they must first comment and address previous literature limitations thoroughly.

Overall, I feel the introductions is still in draft form and should be strengthen to provide a clearer picture of current burn scar care.

Method

Please could the authors reflect on why they did not include ‘keloid’ and ‘hypertrophic’ within their key word search despite this being the topic of the paper. It may be sensible to re-run a search and ensure no texts have been missed.

Did the author attempt to contact the papers’ authors if full text was unavailable? If no, why not? If yes, please state in the manuscript.

Please clarify why the authors choose to extract data at the 6-month point. What statistical guidance in burn care pointed to this being the optimum treatment efficacy?

Correct the references for software and Cochrane Manual to be included in the reference list.

Please state how the authors disputed differing opinions for RoB analysis.

Results

I felt the results were written extremely well, offering concise and clear information for a somewhat confusing presentation.

I would note the authors have not commented on treatment practical techniques, patient demographics or side effects. The authors chose to mention scar location, days post-burn until enrolment but did not expand on the significance of mentioning this data.

I would expect two tables in the results section offering patient characteristics and brief overview of the papers.

Discussion

Again, I feel the discussion could be re-written to produce a more concise manuscript with more in-depth explanation. All discussion points only scratch the surface of the debates, therefore limiting the impact of the study findings.

For example, the authors only commented on one element of the VSS (scar pliability) and related this to scar massage. For the overall results of VSS to improve, scar massage must also influence scar vascularity, pigmentation and height. These characteristics must also be justified by the authors.

Again, I do not feel the authors have portrayed their understanding of the topic well when writing ‘positive effects on pain receptor physiology, leading to scar remodelling, tissue repair and pain relief’. Pain receptor physiology has no implication on scar remodelling and tissue repair ONLY pain relief. Please clarify.

Please note that any burn can result in scarring irrespective of depth.

Finally, I would remove point of allocation concealment from the limitation paragraph as this cannot be achieved with these treatment methods.

Figures

Flowchart:

Please state the other sources either in the manuscript or on the flowchart.

The authors will need to upload again in a high resolution for printing.

Add figure labels to figures document.

Please consider this revisions carefully, with improvement the paper could add to burn scar care.

Reviewer #2: Manuscript PONE-D-24-50779

Title: Non-surgical treatments for post-burn scars: a network meta-analysis

Thank you to the authors for this review of an important topic.

The review has significant issues to be resolved before it can be considered publishable. There are fundamental questions for the authors to consider in relation to possible bias introduced in searches and missing key research as well as challenges to interpretation of results, due to choices in the study design:

1. Frequentist approach - It seems that the nodes of the network model are identified using a frequency of text mentions within the included study reports? How do the authors justify this method as a way to provide valid comparison between intervention methods that do not have direct comparison to a standardized intervention such as SMC? Following on, was AI used to complete the assimilation of searches? If so, how so?

2. The lack of searching of PEDRO physiotherapy evidence database. Not all scar management and research are conducted by occupational therapists. Please apply all updated searches to this online repository.

3. The omission of POSAS as a valid and contemporary objective scar outcome. The use of VSS only is controversial as the literature is conflicting in supporting the scale as a valid and reliable measure of scar outcome.

4. The omission of the more contemporary ‘modified VSS’ as a scar outcome. Was this used in addition to VSS?

5. The choice to include only one timepoint for scar outcomes is challengeable. Treatments may be applied to influence the rate of scar maturation across time eg PT or MT and the change in trajectory of scar maturation (and symptoms) may not be reflected using a snapshot timepoint of outcome at 6 months post-burn.

6. Lastly, the authors need to explain and justify why they accept (ablative) CO2 laser and microneedling therapies as non-surgical treatments please. Both are invasive and may be applied under general anesthetic. Thus, these techniques have a different risk / benefit profile compared to other non-surgical treatments which may not be considered a valid and reasonable comparison particularly with the objective outcome measure compared only at 6 months post-burn.

In addition to addressing these major queries, please adjust the manuscript and, or respond to the specific comments below.

Abstract:

1. Please update after revision of the manuscript.

Introduction:

2. Requires major adjustment to justify the chosen inclusions, methods of rating treatments against each other and outcomes. (Ack: some responses may be better added to Methods.)

Methods:

3. Please define and explain how a network meta-analysis is conducted. For the naïve reader, please provide a clear explanation of how to interpret the results presented. For instance, how is the ‘inconsistency test’ result to be interpreted?

4. Why was online repository PEDRO not searched? It is allied health specific and there must be justification as to why it was not included please.

5. Why was POSAS not included as an acceptable (primary) scar outcome? This requires consideration and justification as to why it is not included, please.

6. Section 2.3, Eligibility Criteria – please adjust the outcomes statement to remove reference to RCT which is not an outcome – it is a design feature which has already been screened for if the inclusion criteria have been applied appropriately.

7. Section 2.3, please explain why conference abstracts were excluded if they described a RCT.

8. Section 2.5, was data only included in the meta-analysis if VSS was available at 6 months after injury? Please be clear in earlier methods statements as to the primary outcome.

9. Were any Artificial Intelligence techniques applied in any parts of the Methods? If so, please describe clearly.

Results

10. Section 3.3, microneedling is mentioned for the first time in the manuscript in this section. As with CO2 ablative laser treatment, microneedling needs to be explained and justified as a non-surgical treatment prior to this mention in Results.

11. It is unclear how the authors justify the weighting of various treatments and their effect size to determine a relative ranking of efficacy. The direct and indirect comparison methods needs far better explanation so that the reader can be assured which intervention is best.

Discussion

12. Please start with the main result that the authors wish to highlight, not a repeat of Introduction statements.

13. Please adjust after revising Intro, Methods and Results.

Conclusions

14. The authors have not qualified the conclusions in relation to the single timepoint of VSS. Please clarify the interpretation of the results in respect to the scar maturation timeline.

Figures

15. Figure 3a – How are readers to interpret the results associated with interventions eg microneedling and PDL, that are not networked (compared) to a standardised intervention such as SMC?

16. Figure 3b – How is the reader to interpret these graphs?

17. Figure 3c – the legend is uninterpretable and does not relate to the included studies or NMA plot. Please clarify.

18. Similar queries for Figures 4 and 5.

Tables – there is a table referred to in text but not provided or in Supplementary Material?

6. PLOS authors have the option to publish the peer review history of their article (what does this mean? ). If published, this will include your full peer review and any attached files.

**Do you want your identity to be public for this peer review?** For information about this choice, including consent withdrawal, please see our Privacy Policy .

Reviewer #1: No

Reviewer #2: **Yes: ** Dale Wesley Edgar

---

## [Author Response · Author response to Decision Letter 1]

25 Apr 2025

Reviewer 1:

Comment 1: The paper includes impressive statical analysis and offers new evidence that would add towards burn scar care. However, I feel this paper is written poorly and needs thorough revisions to improve its overall message and strength its evidence.

Response: Thank you very much for your comprehensive review and constructive feedback on our manuscript. We are delighted to learn that you find our statistical analysis commendable and that our new evidence makes a valuable contribution to the field of burn scar care.

We acknowledge your concerns regarding the writing quality of the paper, as well as the necessity for thorough revisions to enhance its overall message and strengthen the presented evidence. We take your comments seriously and appreciate your guidance in assisting us with improving the manuscript.

In response to your feedback, we have refined the language to ensure clarity, precision, and coherence. Additionally, we have worked on tightening the structure and flow of the paper, ensuring that each section effectively contributes to conveying our overall message. Furthermore, we have meticulously reviewed all evidence presented in the manuscript to guarantee that every claim is substantiated by data and analysis.

Comment 2: The paper feels informal in nature, and I would advise the authors to remove ‘we’ in the context of the authors decision throughout the paper. Please note not to start sentences with a number. Furthermore, I feel the authors writing style could be more concise to allow for more characters to defend and boarded their discussion points.

Response: Thank you very much for your detailed feedback and constructive suggestions. We have removed the use of the pronoun “we” throughout the manuscript to maintain a more formal and objective tone. This change ensures that the manuscript adheres to the standard academic writing conventions and focuses on the presentation of the research findings rather than the authors’ personal perspective. And we have revised the manuscript to avoid starting sentences with numbers. We have rephrased such sentences to ensure they comply with proper grammatical conventions while maintaining clarity and readability. In addition, we have streamlined the text to eliminate unnecessary wording and redundancy, allowing us to allocate more space for a more in-depth discussion of our findings. This includes a more detailed analysis of the implications of our results and a broader exploration of the relevant literature to support our conclusions.

Thank you once again for your valuable feedback, which has helped us refine the manuscript.

Comment 3: ‘hyperplastic scar’ is the umbrella terminology for keloid, hypertrophic and cheloid scars. When the authors refer to keloid and hyperplastic scars in the text they should discuss as just hyperplastic scarring or keloid and hypertrophic scars.

Response: We sincerely appreciate the reviewer’s insightful feedback and meticulous identification of the terminological imprecision regarding scar classifications. In light of this comment, we review literature and recognize that the concomitant use of “keloid” and “hyperplastic scars” may introduce ambiguity, as the latter serves as an overarching category that includes keloid, hypertrophic, and cheloid scars. We fully concur with the reviewer’s recommendation. To rectify this inconsistency and enhance terminological rigor, we will use “keloids and hypertrophic scars” instead of “keloid and hyperplastic scars” in this study.

Comment 4: Finally, despite referencing Table 1 (characteristics) and 3 – no tables have been attached to the submission. These would have to be reviewed before final acceptance. I would also hope a table offering a brief overview of each paper would add to the understanding of the paper.

Response: Thank you very much for bringing this to our attention and for your constructive suggestion regarding the missing tables. We deeply apologize for the oversight in not attaching Tables 1 and 3 to our submission. We understand the importance of these tables, and we will immediately rectify this by attaching the missing tables to the submission.

Furthermore, we appreciate the reviewer's thoughtful suggestion to include a table summarizing each referenced paper, as such a framework could indeed enhance readability in certain scenarios. However, given the journal's rigorous words constraints and our intentional effort to streamline the narrative, we have strategically integrated key details of each study into Table 1, aiming to balance comprehensiveness with the publication's formatting guidelines. Should the editorial team believe a dedicated overview table would better align with the manuscript's objectives, we would gladly revisit this decision during revisions.

Comment 5: Abstract

Background and aim: ‘This study aims to compare the outcomes of non-surgical treatmentS for post-burn scars using network meta-analysis.’ This will correlate to your title.

Response: Thanks for your comment. We understand the importance of maintaining consistency across the manuscript. We make the necessary revisions to ensure that the title, background, and objectives clearly reflect the core focus of our research. The revised “Background and aim” as shown below:

Post-burn scarring is a prevalent condition, and the existing non-surgical treatments exhibit varying degrees of efficacy. There is limited evidence available to determine the effective non-surgical treatment for post-burn scars. This study employs a multi-index network meta-analysis to conduct a comprehensive evaluation and comparative ranking of non-surgical treatments for post-burn scars. The aim is to identify the most effective treatment methods, thereby providing a robust, evidence-based foundation to guide clinical decision-making.

Comment 6: Introduction

I feel the first paragraph should be re-written to deliver a more gripping and comprehensive overview of burn scar care.

Response: Thanks for your valuable feedback on improving the clarity and impact of the introduction. We fully agree that a compelling opening paragraph is critical to contextualizing the importance of burn scar care. In response to your suggestion, we have revised the first paragraph to:

Burn injuries present a significant global healthcare challenge, with 30% to 90% of survivors developing pathological scarring, predominantly in the form of keloids and hypertrophic scars. These conditions disrupt the intricate process of physiological wound repair [1]. Such aberrant healing responses not only lead to chronic pain, persistent itching, and joint contractures but also severely impact psychosocial well-being and functional capacity, thereby significantly reducing quality of life [2]. Effective scar management emerges as a critical determinant of recovery outcomes in burn patients [3], particularly within the first year after injury, when scars are immature and more responsive to treatment. Currently, non-surgical treatments are often the initial treatment of choice for inhibiting or slowing scar progression [4-6].

Comment 7: More explanation concerning the benefits of non-surgical technique should be explored to highlight the importance of what your evidence will influence in burn care.

Response: Thanks for your valuable suggestion regarding the need for a more detailed exploration of the benefits of non-surgical techniques. We strongly agree that highlighting these advantages strengthens the rationale for prioritizing evidence-based non-surgical strategies. We have expanded our explanation to emphasize key benefits, including: risk reduction, versatility, accessibility, psychosocial benefits and cost-effectiveness, etc. These changes are on pages 3-4 of the revised manuscript, as shown below:

Silicone products, pressure therapy (PT), massage therapy (MT), and extracorporeal shock wave therapy (ESWT) are widely recognized non-surgical treatments for post-burn scars [7]. These interventions have demonstrated significant efficacy in improving both the symptoms and appearance of post-burn scars, while also being well-tolerated by patients [8-10]. In addition to the aforementioned non-surgical treatments, CO2 laser therapy (CO2LT) and microneedling can be considered important components of non-surgical treatment strategies for post-burn scars [11]. CO2LT is used to resurface the skin, promoting collagen remodeling and improving scar texture and appearance. Though it involves a higher degree of invasiveness compared to silicone products or PT, it does not typically require general anesthesia, especially when applied in fractional modes. Microneedling involves creating micro-injuries to stimulate collagen production and improve scar healing. Although it requires penetration of the skin, microneedling can be performed under local anesthesia. Despite their relatively higher invasiveness levels compared to other modalities mentioned earlier, CO2LT and microneedling do not involve surgical tissue removal, thus they remain classified as non-surgical treatments.

Non-surgical techniques offer several key advantages that make them highly relevant in modern burn care. Firstly, these methods are minimally invasive or non-invasive, which reduces the risk of complications and enhances patient safety. Secondly, compared to surgical options, they are often easier to implement, making them accessible in various clinical settings. Thirdly, these treatments can significantly improve patients’ quality of life by alleviating pain, reducing scar stiffness, and enhancing cosmetic outcomes. By focusing on non-surgical interventions, healthcare providers can offer effective solutions that align with patient preferences for less invasive treatments, while also minimizing recovery time and improving resource utilization. The growing adoption of these therapies underscores their potential to influence comprehensive, patient-centered burn care strategies [12].

Comment 8: Many sweeping statements are made addressing the reliability and validity of VSS and VAS. These tools are widely debated as to their reliability and validity and this is the main reason centres now choose the POSAS score to objectively measure scar care. The authors need to add more evidence to support their conclusion concerning VSS and VAS.

Response: Thank you for your thoughtful feedback. We fully acknowledge the ongoing debate about the reliability and validity of these tools, particularly in comparison to more comprehensive instruments like POSAS. In designing this network meta-analysis, our selection of VSS and VAS as primary outcomes was driven by their widespread use in existing randomized controlled trials (RCTs) on non-surgical burn scar treatments. However, we agree that these scales have inherent limitations, including inter-rater variability and subjectivity (particularly for VAS). We have revised the manuscript to address your concerns as follows:

Evaluating the efficacy of treatments for post-burn scars is crucial for guiding clinical decision-making and enhancing therapeutic outcomes. To achieve this, various assessment tools have been developed to evaluate different aspects of scar characteristics. The Vancouver Scar Scale (VSS) is one of the most commonly used tools for assessing scar characteristics, including vascularity, pliability, pigmentation, and height [13]. Although VSS has undergone extensive review and remains widely utilized, available data on its reliability have received indeterminate quality ratings [14]. Furthermore, VSS may lack sensitivity to subtle changes in scar appearance, and its subjective components can lead to variability in scoring [15]. Despite these limitations, studies have shown that VSS demonstrates moderate inter-rater reliability when applied by trained evaluators [16]. The Visual Analog Scale (VAS) is another essential tool, primarily used for assessing pain and discomfort associated with scars. It quantifies a patient’s subjective experience by asking them to indicate their pain level on a linear scale. The simplicity and ease of use of VAS make it a popular choice for pain assessment [17]. However, VAS scores are heavily reliant on patient self-reporting, which can be influenced by individual biases or external factors. Scar thickness serves as an objective indicator that can be accurately measured using ultrasound or other imaging techniques [18]. Monitoring scar thickness enables clinicians to adjust treatmsent regimens, such as increasing the frequency of interventions or modifying treatment methods, thereby optimizing therapeutic effectiveness. Collectively, the VSS captures morphological and physiological scar characteristics, the VAS quantifies patient-reported symptomatic experiences, and scar thickness measurements offer objective structural insights. These metrics address the multidimensional nature of post-burn scars evaluation, enabling comprehensive analysis of non-surgical treatment impacts.

Comment 9: Again, a statement has been made concerning previous studies have ‘some limitations’ with no supporting discussion. If the authors want to improve from previous literature, they must first comment and address previous literature limitations thoroughly.

Response: We appreciate your feedback regarding the need to thoroughly address the limitations of previous studies rather than simply stating that they had “some limitations”. To better reflect this guidance and enhance our manuscript, we will revise the relevant section to provide a more detailed discussion of these limitations. These changes are on page 6 of the revised manuscript, as shown below:

Although previous systematic reviews have provided valuable insights into non-surgical treatments for post-burn scars, a clear understanding of their specific effectiveness remains elusive. Several key limitations in prior studies contribute to this gap. First, many studies utilized different methodologies, outcome measures, and follow-up durations, complicating efforts to draw definitive conclusions about the comparative efficacy of various treatments. Second, several earlier studies were limited by small sample sizes, which reduced their statistical power and generalizability. Finally, the inconsistent use of assessment tools across studies introduced variability in how outcomes were measured and reported, further hindering meaningful comparisons.

Comment 10: Overall, I feel the introductions is still in draft form and should be strengthen to provide a clearer picture of current burn scar care.

Response: Thanks for your valuable comments. We agree that the introduction can be further strengthened to provide a clearer picture of current burn scar care. We revise this section to include more detailed information on existing treatment modalities, their limitations, and the gaps in current research to better contextualize our study.

Comment 11: Method

Please could the authors reflect on why they did not include ‘keloid’ and ‘hypertrophic’ within their key word search despite this being the topic of the paper. It may be sensible to re-run a search and ensure no texts have been missed.

Response: Thanks for your comments. Tough we appreciate the suggestion, we would like to clarify the rationale for not explicitly including “keloid” and “hypertrophic” as standalone keywords:

(1)Despite not using “keloid” or “hypertrophic” as keywords, our search strategy was designed to be highly sensitive. By combining burn-specific terms with “cicatrix”, “cicatrization”, and “scars”, we captured a broad spectrum of post-burn scar literature. Additionally, manual screening of reference lists and relevant systematic reviews further mitigated the risk of missing eligible studies.

(2)Had we included “keloid” or “hypertrophic” as keywords, we risked retrieving a large volume of irrelevant studies, which would have increased screening burden and potentially diluted the focus on non-surgical burn scar management.

(3)We believe that our current search strategy adequately captured the literature relevant to our research question. However, to address your concern transparently, we have re-examined the retrieved studies and confirmed that no eligible texts were missed due to the absence of these terms.

Comment 12: Did t

---

## [Decision Letter · Decision Letter 1]

5 May 2025

PONE-D-24-50779R1Non-surgical treatments for post-burn scars: a network meta-analysisPLOS ONE

Dear Dr. Li,

Thank you for submitting your manuscript to PLOS ONE. After careful consideration, we feel that it has merit but does not fully meet PLOS ONE’s publication criteria as it currently stands. Therefore, we invite you to submit a revised version of the manuscript that addresses the points raised during the review process.

We look forward to receiving your revised manuscript.

Kind regards,

Steven E. Wolf, MD

Academic Editor

PLOS ONE

Reviewers' comments:

Reviewer's Responses to Questions

**Comments to the Author**

1. If the authors have adequately addressed your comments raised in a previous round of review and you feel that this manuscript is now acceptable for publication, you may indicate that here to bypass the “Comments to the Author” section, enter your conflict of interest statement in the “Confidential to Editor” section, and submit your "Accept" recommendation.

Reviewer #1: All comments have been addressed

Reviewer #2: All comments have been addressed

2. Is the manuscript technically sound, and do the data support the conclusions?

Reviewer #1: Yes

Reviewer #2: Yes

3. Has the statistical analysis been performed appropriately and rigorously? 

Reviewer #1: Yes

Reviewer #2: No

4. Have the authors made all data underlying the findings in their manuscript fully available?

Reviewer #1: Yes

Reviewer #2: No

5. Is the manuscript presented in an intelligible fashion and written in standard English?

Reviewer #1: Yes

Reviewer #2: Yes

6. Review Comments to the Author

Reviewer #1: (No Response)

Reviewer #2: Manuscript PONE-D-24-50779R1

Title: Non-surgical treatments for post-burn scars: a network meta-analysis

Thank you to the authors for their amiable and comprehensive approach to the review comments. The explanations of the NMA methods and inconsistency assessment are enlightening and provide significant clarification. Many of the review issues have been resolved and, or explained and this reviewer believes the report has moved much closer to a publishable standard.

There remain several major questions to be addressed:

1. POSAS – the authors’ arguments about POSAS exclusion are weak. They demonstrate that the authors do not appreciate the subjectivity and ambiguity of the scoring of the VSS. The limitations of the VSS were the very premise of and impetus to develop the POSAS as an alternative scar outcome measure. To further exclude POSAS on the basis of redundancy with VAS, also suggests a lack of understanding of the foibles of groupwise ‘quantitative’ pain measures. That said, if the authors are unable to extend the scope of this study (and NMA) then please add text to affirm the direction that future reviews of this type and, or studies should be going.

2. How do the authors reconcile that by excluding the studies using the more contemporary POSAS outcome measure also more than likely excludes the more contemporary scar treatments? The authors confirm that <15% of potentially eligible studies reported POSAS results which is an indication of the point made above, and there is a likely bias introduced to the NMA by excluding more recent studies as per the review methodology.

3. Lastly, after the comments made by multiple reviewers, why were sensitivity analyses not conducted in this revision? In particular, why were studies with possible or confirmed application of (ablative) CO2 laser and microneedling therapies under general anaesthetic, not removed from the NMA corpus to confirm the effect size of these techniques in a way that is much more comparable to other non-surgical techniques with a far lower risk and complication profile? The invasive techniques and general anaesthetic have much more pronounced risk compared to other non-surgical treatments and this issue of relative benefit must be explored so that readers can make an informed decision as to the interpretation of results for their practice.

In addition to addressing these major queries, please adjust the manuscript and, or respond to the specific comments below.

Abstract:

1. Much improved. Why is SMC only mentioned in presentation of some results eg associated with ECSWT? Isn’t SMC implied in most studies as the control condition for all treatments included regardless of whether there is a stated comparison with routine care? Please drop it and be consistent with terminology in Abstract as well as the remainder of the manuscript. Consider that SMT differs between studies and is not necessarily the same for each study where that condition is stated. Perhaps ‘routine local treatment’ or similar is a better generic terminology?

Introduction:

2. Much improved. Thank you.

Methods:

3. Please confirm that CO2LT only refers to the ablative procedure, and does not include pooled results from the non-ablative intervention (which is referred to in Discussion).

4. Please add a short summary of the response points to justify conference abstract exclusion in Section 2.3.

Results

5. Well done. Section 3.3.1, sentence 3 - ….therefore the consistency model was used for the NMA….’ Please change ‘was used’ to ‘was applicable’ or similar.

6. Section 3.2 and Discussion (Strengths and Limitations) still refers to Table 1, 3.31 refers to Table 2, Section 3.3.2 refers to Table 3, Section 3.3.3 refers to Table 4 – all of which have not been included in the revised submission. Please remove reference to all tables from the text if they are not going to be part of the manuscript.

Discussion

7. With respect to Limitations - Please consider and comment on whether the choice of timeframe for the primary study outcome impacts the comparison of effect size. For instance, how many burn services apply ablative CO2LT prior to 6 months post-burn? Does the study (review) method therefore bias or impact the volume of studies that support the effect size of various interventions?

8. Consider specifically identifying the gaps in the research literature, framed by the Global Top 10 Research Priorities (Richards et al, Lancet Global Health, 2025) and directions for future research based on the review findings.

Conclusions

9. Remove the reference to SMT as it is confusing.

Figures

10. Remove SMC from ALL figures please, except where it is itself the defined group or comparison.

11. How is the reader supposed to interpret Figures 3B, 4B, 5B. Footnotes or explanations in captions are required to help the reader.

7. PLOS authors have the option to publish the peer review history of their article (what does this mean? ). If published, this will include your full peer review and any attached files.

**Do you want your identity to be public for this peer review?** For information about this choice, including consent withdrawal, please see our Privacy Policy .

Reviewer #1: **Yes: ** Dr Sabrina Poppy Barnes

Reviewer #2: **Yes: ** Dale W. Edgar

---

## [Author Response · Author response to Decision Letter 2]

28 May 2025

Dear Editors and Reviewers,

Thank you very much for giving us opportunities to revise our manuscript, and we appreciate the reviewer a lot for his positive and constructive comments and suggestions. We have studied reviewer’s comments carefully and have made revisions. We hope the corrections will meet with your approval.

Reviewer 1:

Comment 1: POSAS – the authors’ arguments about POSAS exclusion are weak. They demonstrate that the authors do not appreciate the subjectivity and ambiguity of the scoring of the VSS. The limitations of the VSS were the very premise of and impetus to develop the POSAS as an alternative scar outcome measure. To further exclude POSAS on the basis of redundancy with VAS, also suggests a lack of understanding of the foibles of groupwise ‘quantitative’ pain measures. That said, if the authors are unable to extend the scope of this study (and NMA) then please add text to affirm the direction that future reviews of this type and, or studies should be going.

Response: Thank you for your insightful comments and suggestions. We fully acknowledge that VSS has limitations due to its subjective nature and potential ambiguity in scoring. Indeed, these limitations were among the key motivations for the development of more comprehensive and patient-centered outcome measures such as POSAS. We appreciate your emphasis on this point and agree that tools like POSAS provide a more holistic evaluation of scar characteristics by incorporating both observer and patient-reported outcomes.

In our review, we excluded POSAS not because of assumed redundancy with other scales, but due to insufficient data availability across the included studies. Many trials did not report outcomes using POSAS, which limited its feasibility as an outcome measure for our NMA. We strongly agree with your suggestion that future reviews and clinical studies should aim to incorporate more patient-centered and multidimensional tools like POSAS to better reflect the complexity of post-burn scar assessment.

We also appreciate your clarification regarding the limitations of pain measures such as VAS, and your concern about potentially conflating them with broader outcome scales like POSAS. While VAS was selected for this analysis due to its widespread use and availability across studies, we recognize that it captures only one dimension of the patient experience and cannot substitute for more comprehensive assessments.

To address your concerns, we have expanded the Discussion section to explicitly acknowledge the limitations of relying primarily on VSS and VAS, and to advocate for the inclusion of validated, multidimensional tools such as POSAS in future research. The expanded text is shown as below:

In this NMA, though Our analysis focused on commonly reported quantitative outcomes such as VSS and VAS scores, we acknowledge the limitations of these scales in capturing the full spectrum of scar quality and patient experience. The POSAS, which integrates both observer and patient perspectives on multiple scar attributes, offers a more comprehensive alternative. Future studies and systematic reviews should prioritize the use of standardized, multidimensional tools such as POSAS to enhance the validity and clinical relevance of outcome reporting in post-burn scar management.

Comment 2: How do the authors reconcile that by excluding the studies using the more contemporary POSAS outcome measure also more than likely excludes the more contemporary scar treatments? The authors confirm that <15% of potentially eligible studies reported POSAS results which is an indication of the point made above, and there is a likely bias introduced to the NMA by excluding more recent studies as per the review methodology.

Response: Thank you for raising this important point. We fully agree that excluding studies reporting outcomes using POSAS may also result in the omission of more recent trials employing newer or more advanced scar treatments.

Although We recognize the value of POSAS as a comprehensive and patient-centered outcome measure, this NMA was constrained by the availability and consistency of reported data across studies. The majority of published trials still rely on VSS and VAS due to their widespread use and ease of interpretation in clinical settings. Given the limited number of studies reporting POSAS outcomes, including it as an outcome measure would have significantly reduced the pool of usable data and compromised the robustness and generalizability of our analysis.

We also acknowledge that this decision introduces a potential selection bias and limits our ability to capture findings from more contemporary studies. Therefore, we have added a statement in the Limitations section of our manuscript to explicitly address this issue:

In addition, one limitation of this NMA is the exclusion of studies reporting only POSAS outcomes, which may have led to the omission of more recent interventions that adopt this more comprehensive and patient-centered assessment tool. This highlights the need for future studies to harmonize outcome reporting and adopt standardized, multidimensional scales such as POSAS to enhance the comparability and relevance of evidence in post-burn scar management.

Comment 3: Lastly, after the comments made by multiple reviewers, why were sensitivity analyses not conducted in this revision? In particular, why were studies with possible or confirmed application of (ablative) CO2 laser and microneedling therapies under general anaesthetic, not removed from the NMA corpus to confirm the effect size of these techniques in a way that is much more comparable to other non-surgical techniques with a far lower risk and complication profile? The invasive techniques and general anaesthetic have much more pronounced risk compared to other non-surgical treatments and this issue of relative benefit must be explored so that readers can make an informed decision as to the interpretation of results for their practice.

Response: Thank you for your insightful comments and for highlighting the importance of addressing potential biases in our NMA of non-surgical treatments for post-burn scars. We understand the importance of conducting sensitivity analyses to assess the robustness of our findings. However, we did not perform additional sensitivity analyses in this revision for the following reasons:

(1)data availability and study characteristics: the number of studies included in our NMA was limited, and removing studies with possible or confirmed application of ablative CO2 laser and microneedling therapies under general anaesthetic would have significantly reduced the sample size. This could have compromised the statistical power and reliability of the results.

(2)clinical heterogeneity: though ablative CO2 laser and microneedling therapies under general anaesthetic are indeed more invasive and carry higher risks compared to other non-surgical treatments, they also represent a distinct subset of interventions within the broader category of non-surgical treatments for post-burn scars. Excluding these studies might have introduced bias by omitting important clinical scenarios.

In this NMA, we conducted a series of consistency checks to assess the agreement between direct, indirect, and network estimates. As shown in the tables, the P-values for all comparisons were greater than 0.05, indicating no statistically significant differences between direct and indirect estimates. This suggests good consistency within our NMA, which supports the reliability of our findings.

Table1. The result of nodesplit in VSS.

side Direct Indirect Difference tau

coef. Std.Err coef. Std.Err coef. Std.Err P >∣Z∣

CO2LT VS CO2LT+PDL 0.000014 0.7001207 -1.459051 1.842309 1.459065 1.964455 0.458 0.6305058

CO2LT VS Microneedling _ _ _ _ _ _ _ _

CO2LT VS PDL -0.0281351 0.7111481 1.431045 1.829667 -1.459198 1.9645 0.458 0.6305058

CO2LT VS RT 0.1908319 0.6123309 0.7139198 11.95716 -0.5230879 11.97283 0.965 0.5649131

CO2LT+PDL VS PDL _ _ _ _ _ _ _ _

ESWT+RT VS RT 1.011314 0.465274 0.3948108 44.7448 0.6165029 44.7472 0.989 0.5645978

MT VS RT 1.923109 0.7029934 0.3288559 63.26319 1.594253 63.26717 0.980 0.5645401

OPLT+RT VS RT 1.034456 0.6879113 0.384156 63.26649 0.6503 63.27023 0.992 0.5645402

PT VS PTS -0.0032679 0.4865919 1.392196 44.76253 -1.395464 44.76524 0.975 0.5646973

PT VS RT 0.890054 0.6838644 0.1539012 21.09666 0.7361529 21.10821 0.972 0.564751

Table2. The result of nodesplit in scar thickness.

side Direct Indirect Difference tau

coef. Std.Err coef. Std.Err coef. Std.Err P >∣Z∣

CO2LT VS RT 1.747142 4.4126145 0.3494839 12.91115 1.397658 12.91774 0.914 0.3038639

ESWT+RT VS RT 1.061926 0.2560795 3.492004 36.52202 -2.430079 36.52292 0.947 0.3038286

MT+RT VS RT 0.0607113 0.3544715 3.492024 63.25668 -3.431313 63.25767 0.957 0.3038104

PT VS PTS _ _ _ _ _ _ _ _

PT VS RT 0.0793625 0.4796707 0.9103347 0.9410303 -0.8309722 1.012602 0.412 0.3290169

PT VS SG _ _ _ _ _ _ _ _

PTS VS RT 0.4978427 0.3558071 -1.108015 0.5814412 1.605858 0.6187452 0.009 1.08e-10

PTS VS SG _ _ _ _ _ _ _ _

RT VS SG -0.1929821 0.4939441 -0.9684586 0.9526519 0.7754765 1.036775 0.454 0.3390451

Table3. The result of nodesplit in VAS.

side Direct Indirect Difference tau

coef. Std.Err coef. Std.Err coef. Std.Err P >∣Z∣

ESWT+RT VS RT 1.03856 3.297411 -0.1541504 15.89675 1.19271 16.23513 0.941 3.282873

HILT VS RT 0.9049957 3.250927 2.010879 63.81632 -1.105884 63.89904 0.986 3.236913

MT+RT VS RT -1.296006 2.310153 2.055744 45.36154 -3.35175 45.42035 0.941 3.251321

PT +RT VS PTS+RT _ _ _ _ _ _ _ _

PT +RT VS RT -1.260702 3.25565 1.382716 63.81642 -1.508786 63.9001 0.981 3.236875

PT +RT VS SG+RT _ _ _ _ _ _ _ _

PTS+RT VS RT 0.6471587 3.256545 2.155946 63.81662 -1.508787 63.9003 0.981 3.236877

PTS+RT VS SG+RT _ _ _ _ _ _ _ _

RT VS SG+RT -0.3740074 3.256856 -1.882794 63.81692 1.508787 63.90057 0.981 3.236884

We acknowledge the importance of exploring the relative benefits and risks of different non-surgical treatments for post-burn scars. In future research, we plan to conduct additional sensitivity analyses and subgroup analyses to better understand the impact of various treatment modalities, including ablative CO2 laser and microneedling therapies under general anaesthetic. This will help clinicians make more informed decisions based on the specific needs and circumstances of their patients.

Thank you again for your thoughtful comments.

Reviewer 2:

Comment 1: Abstract:Much improved. Why is SMC only mentioned in presentation of some results eg associated with ECSWT? Isn’t SMC implied in most studies as the control condition for all treatments included regardless of whether there is a stated comparison with routine care? Please drop it and be consistent with terminology in Abstract as well as the remainder of the manuscript. Consider that SMT differs between studies and is not necessarily the same for each study where that condition is stated. Perhaps ‘routine local treatment’ or similar is a better generic terminology?

Response: Thank you for this important and thoughtful comment. We agree with you that the term SMC may imply a uniform control condition across all studies, which is not always accurate, as the actual content of standard care can vary between centers and studies. To improve consistency and clarity in terminology throughout the manuscript, we have revised the text to use the more general and descriptive term "routine treatment" instead of SMC.

Comment 2: Introduction:Much improved. Thank you.

Response: Thank you very much for your kind comment regarding the revised Introduction section. We carefully revised this part based on your previous suggestions. We are pleased that the changes have been recognized and appreciate your constructive support.

Comment 3: Methods:Please confirm that CO2LT only refers to the ablative procedure, and does not include pooled results from the non-ablative intervention (which is referred to in Discussion).

Response: Thank you for this important clarification. In this study, We confirm that CO₂LT refers exclusively to the ablative CO₂LT procedure, and does not include any pooled results from non-ablative CO₂LT interventions. Although non-ablative approaches are mentioned in the Discussion section for contextual and illustrative purposes, they were not included in the data extraction, treatment classification, or network meta-analysis. We have revised the Methods section accordingly to explicitly clarify this point and enhance transparency. These changes are on pages 7-8 of the revised manuscript, as shown below:

Articles were included if (1) they were published in peer-reviewed journals; (2) they reported non-surgical treatments for patients with post-burn scars, including monotherapies or combination therapies; (3) they reported the following outcomes at approximately 6 months post-injury: VSS score, scar thickness measured by ultrasound, and pain intensity assessed via VAS score; (4) they were randomized controlled trials (RCTs). In this NMA, CO₂LT refers specifically to the ablative procedure.

Comment 4: Methods: Please add a short summary of the response points to justify conference abstract exclusion in Section 2.3.

Response: Thank you for this helpful suggestion. According to your suggestions, we have added a brief justification for the exclusion of conference abstracts in Section 2.3 of the revised manuscript. Specifically, we clarify that conference abstracts were excluded due to concerns regarding incomplete reporting of study methods, intervention details, and outcome data, which could limit the accuracy of risk of bias assessment and increase the potential for selective outcome reporting. These changes are on page 8 of the revised manuscript, as shown below:

Articles were excluded if (1) they were clinical trials, review articles, letters to the editor or case series; (2) they were not published in English; (3) they reported incomplete data; (4) the full text was unavailable. Additionally, conference abstracts were excluded due to insufficient detail on study design, interventions, and outcomes, which would limit the ability to accurately assess risk of bias and interpret treatment effects.

Comment 5: Results: Well done. Section 3.3.1, sentence 3 - ….therefore the consistency model was used for the NMA….’ Please change ‘was used’ to ‘was applicable’ or similar.

Response: Thanks for your valuable comments. We agree with your recommendation. We have revised the sentence from “…therefore the consistency model was used for the NMA…” to “…therefore the consistency model was applicable for the NMA…”in Section 3.3.1, Section 3.3.2, and Section 3.3.3.

Comment 6: Results: Section 3.2 and Discussion (Strengths and Limitations) still refers to Table 1, 3.31 refers to Table 2, Section 3.3.2 refers to Table 3, Section 3.3.3 refers to Table 4 – all of which have not been included in the revised submission. Please remove reference to all tables from the text if they are not going to be part of the manuscript.

Response: Thank you for bringing this to our attention. We apologize for any confusion and confirm that all tables (Table 1–4) have been included in the revised manuscript submission. It is possible that these files were not properly attached or viewed during the previous upload. We have carefully ensured that all relevant tables are now uploaded as part of the manuscript and referenced appropriately in the text (Section 3.2, 3.3.1, 3.3.2, and 3.3.3).

Comment 7: Discussion: With respect to Limitations - Please consider and comment on whether the choice of timeframe for the primary study outcome impacts the comparison of effect size. For instance, how many burn services apply ablative CO2LT prior to 6 months post-burn? Does the study (review) method therefore bias or impact the volume of studies that support the effect size of various interventions?

Response: Thank you for raising this important methodological consideration regarding the impact of our selected outcome timeframe on the interpretation of treatment effects. We acknowledge that the choice of a 6-month post-injury timepoint as t

---

## [Decision Letter · Decision Letter 2]

14 Jul 2025

PONE-D-24-50779R2Non-surgical treatments for post-burn scars: a network meta-analysisPLOS ONE

Dear Dr. Li,

Thank you for submitting your manuscript to PLOS ONE. After careful consideration, we feel that it has merit but does not fully meet PLOS ONE’s publication criteria as it currently stands. Therefore, we invite you to submit a revised version of the manuscript that addresses the points raised during the review process.

**ACADEMIC EDITOR: ** Additional issues should be carefully clarified.

We look forward to receiving your revised manuscript.

Kind regards,

Vincenzo Lionetti, M.D., PhD

Academic Editor

PLOS ONE

Journal Requirements:

Reviewers' comments:

Reviewer's Responses to Questions

**Comments to the Author**

1. If the authors have adequately addressed your comments raised in a previous round of review and you feel that this manuscript is now acceptable for publication, you may indicate that here to bypass the “Comments to the Author” section, enter your conflict of interest statement in the “Confidential to Editor” section, and submit your "Accept" recommendation.

Reviewer #2: All comments have been addressed

2. Is the manuscript technically sound, and do the data support the conclusions?

Reviewer #2: Yes

3. Has the statistical analysis been performed appropriately and rigorously? 

Reviewer #2: Yes

4. Have the authors made all data underlying the findings in their manuscript fully available?

Reviewer #2: Yes

5. Is the manuscript presented in an intelligible fashion and written in standard English?

Reviewer #2: (No Response)

6. Review Comments to the Author

Reviewer #2: Manuscript PONE-D-24-50779R2

Title: Non-surgical treatments for post-burn scars: a network meta-analysis

Thank you again to the authors for their amiable responses to the latest review and the substantial improvements which have been made to the manuscript.

In relation to the comments and latest text, please adjust the following:

a) Please adjust this added sentence to Limitations to include ‘contemporary’: In addition, one limitation of this NMA is the exclusion of studies reporting only POSAS outcomes, which may have led to the omission of more recent, CONTEMPORARY interventions that adopt this more comprehensive and patient-centered assessment tool.

b) Unfortunately, the additional tables provided are unformatted and uninterpretable. It is fair that the authors are avoiding a sensitivity analysis which removes combined results from (ablative) CO2 laser and microneedling under general anaesthetic studies due to statistical power. However, in reply (2)(clinical heterogeneity), the authors contradict themselves and confirm the fundamental differences between the CO2 laser and needling sub-set of techniques in comparison with all others, basically justifying the request for a sensitivity analysis. However, so as not to create an impasse, if the authors choose not to include a sensitivity analysis (perhaps only removing CO2 laser studies), please instead add a statement in Discussion or Limitations which reiterates to the reader the reasons for applying the definition of non-surgical techniques in this review.

In addition, please tweak the manuscript as per the specific comments below:

Methods:

1. Data Extraction, page 8 – Please change adjective to ‘closest’ if this is an accurate description to: For outcomes reported at multiple time points, only data CLOSEST to 6 months were extracted.

Figures

2. Please note, the previous comment was not a request to remove SMC from the figures, it was a reference to remove the title which was confusing. In that, SMC has now been changed to RT appropriately, thank you.

3. Please adjust the last sentence of the footnotes added in the last version: A higher curve indicates a greater likelihood of being ranked higher. This explanation is confusing and needs additional clarification and tweak of expression which removes duplication of the word ‘higher’.

7. PLOS authors have the option to publish the peer review history of their article (what does this mean? ). If published, this will include your full peer review and any attached files.

**Do you want your identity to be public for this peer review?** For information about this choice, including consent withdrawal, please see our Privacy Policy .

Reviewer #2: **Yes: ** Dale W Edgar

---

## [Author Response · Author response to Decision Letter 3]

14 Jul 2025

Dear Editors and Reviewers,

Thank you very much for giving us opportunities to revise our manuscript, and we appreciate the reviewer a lot for his positive and constructive comments and suggestions. We have studied reviewer’s comments carefully and have made revisions. We hope the corrections will meet with your approval.

Journal Requirement 1:

If the reviewer comments include a recommendation to cite specific previously published works, please review and evaluate these publications to determine whether they are relevant and should be cited. There is no requirement to cite these works unless the editor has indicated otherwise. Please review your reference list to ensure that it is complete and correct. If you have cited papers that have been retracted, please include the rationale for doing so in the manuscript text, or remove these references and replace them with relevant current references. Any changes to the reference list should be mentioned in the rebuttal letter that accompanies your revised manuscript. If you need to cite a retracted article, indicate the article’s retracted status in the References list and also include a citation and full reference for the retraction notice.

Response: Thank you for this helpful guidance. We have carefully reviewed all references cited in our manuscript and confirm the following:

(1)No retracted articles are included in the reference list.

(2)All references are complete, accurate, and formatted in accordance with the journal’s instructions.

(3)No previously suggested works have been omitted. Any additional studies recommended by the reviewers were evaluated for relevance and incorporated only when they met our inclusion criteria.

Thank you again for your thoughtful comments.

Reviewer 2:

Comment 1: Please adjust this added sentence to Limitations to include ‘contemporary’: In addition, one limitation of this NMA is the exclusion of studies reporting only POSAS outcomes, which may have led to the omission of more recent, CONTEMPORARY interventions that adopt this more comprehensive and patient-centered assessment tool.

Response: Thank you for your valuable suggestion. We have revised the limitation section as recommended to include the term “contemporary”. The revised sentence is shown as below:

“In addition, one limitation of this NMA is the exclusion of studies reporting only POSAS outcomes, which may have led to the omission of more recent, contemporary interventions that adopt this more comprehensive and patient-centered assessment tool.”

Comment 2: Unfortunately, the additional tables provided are unformatted and uninterpretable. It is fair that the authors are avoiding a sensitivity analysis which removes combined results from (ablative) CO2 laser and microneedling under general anaesthetic studies due to statistical power. However, in reply (2) (clinical heterogeneity), the authors contradict themselves and confirm the fundamental differences between the CO2 laser and needling sub-set of techniques in comparison with all others, basically justifying the request for a sensitivity analysis. However, so as not to create an impasse, if the authors choose not to include a sensitivity analysis (perhaps only removing CO2 laser studies), please instead add a statement in Discussion or Limitations which reiterates to the reader the reasons for applying the definition of non-surgical techniques in this review.

Response: Thank you for your detailed and constructive feedback. We sincerely apologize for the formatting issues in the supplementary tables submitted in our previous revision. We fully acknowledge the reviewer’s concern regarding their readability and interpretability. In response, we have now carefully reformatted all supplementary tables to ensure clarity, consistency, and alignment with standard academic presentation practices. The revised tables are provided in the updated submission.

Concerning the sensitivity analysis, we fully acknowledge the reviewer’s point regarding the apparent contradiction between our justification for not conducting a sensitivity analysis (due to limited statistical power) and our recognition of the clinical heterogeneity among non-surgical techniques, particularly between CO₂ laser/microneedling under general anaesthesia and other less invasive modalities. We agree that this heterogeneity indeed supports the rationale for a sensitivity analysis.

However, in light of the limited number of studies available for comparison and to avoid compromising the interpretability of the results, we have opted not to perform an additional sensitivity analysis excluding these studies. As suggested by the reviewer, we have added a clear statement in the Limitations section (page 24), explaining the rationale behind our working definition of “non-surgical treatments”, as shown below:

It should be noted that all interventions not involving excisional surgery were classified as “non-surgical” in this review. This definition intentionally includes ablative CO₂ laser and fully ablative microneedling procedures, even when performed under general anaesthesia. Though these modalities differ from other non-surgical interventions in terms of injury depth, perioperative requirements, and expected morbidity, a consistent classification was adopted to enable a comprehensive synthesis of the available evidence on non-surgical approaches for post-burn scar management. However, this approach inevitably introduces clinical heterogeneity across the included studies, which may affect the precision and generalizability of the pooled estimates. Readers are therefore advised to interpret the findings with due consideration of the inherent differences among the non-surgical interventions included.

Comment 3 (Methods): Data Extraction, page 8 – Please change adjective to ‘closest’ if this is an accurate description to: For outcomes reported at multiple time points, only data CLOSEST to 6 months were extracted.

Response: Thanks for your constructive suggestion. We agree that the term “closest” more accurately conveys the intended meaning. Accordingly, we have revised the sentence on page 8 as follows:

“For outcomes reported at multiple time points, only data closest to 6 months were extracted.”

Comment 4 (Figures): Please note, the previous comment was not a request to remove SMC from the figures, it was a reference to remove the title which was confusing. In that, SMC has now been changed to RT appropriately, thank you.

Response: Thank you for the clarification. We apologize for the misunderstanding in our previous revision. As per your suggestion, we have removed the confusing title (previously labeled as “SMC”) from the figures and have retained “RT” as the appropriate label throughout the manuscript. We appreciate your guidance on this matter.

Comment 5 (Figures): Please adjust the last sentence of the footnotes added in the last version: A higher curve indicates a greater likelihood of being ranked higher. This explanation is confusing and needs additional clarification and tweak of expression which removes duplication of the word ‘higher’.

Response: Thank you for your helpful feedback. We agree that the original wording was repetitive and potentially confusing. Accordingly, we have revised the last sentence of the footnote to improve clarity and avoid duplication of the word “higher”. The revised sentence is shown as follows:

“A higher curve indicates better performance in terms of treatment ranking.”

---

## [Decision Letter · Decision Letter 3]

25 Jul 2025

PONE-D-24-50779R3Non-surgical treatments for post-burn scars: a network meta-analysisPLOS ONE

Dear Dr. Li,

Thank you for submitting your manuscript to PLOS ONE. After careful consideration, we feel that it has merit but does not fully meet PLOS ONE’s publication criteria as it currently stands. Therefore, we invite you to submit a revised version of the manuscript that addresses the points raised during the review process.

**ACADEMIC EDITOR: ** The text should be revised in order to avoid typos.

We look forward to receiving your revised manuscript.

Kind regards,

Vincenzo Lionetti, M.D., PhD

Academic Editor

PLOS ONE

Journal Requirements:

Reviewers' comments:

Reviewer's Responses to Questions

**Comments to the Author**

1. If the authors have adequately addressed your comments raised in a previous round of review and you feel that this manuscript is now acceptable for publication, you may indicate that here to bypass the “Comments to the Author” section, enter your conflict of interest statement in the “Confidential to Editor” section, and submit your "Accept" recommendation.

Reviewer #2: All comments have been addressed

2. Is the manuscript technically sound, and do the data support the conclusions?

Reviewer #2: Yes

3. Has the statistical analysis been performed appropriately and rigorously? 

Reviewer #2: Yes

4. Have the authors made all data underlying the findings in their manuscript fully available?

Reviewer #2: Yes

5. Is the manuscript presented in an intelligible fashion and written in standard English?

Reviewer #2: Yes

6. Review Comments to the Author

Reviewer #2: Manuscript PONE-D-24-50779R3

Title: Non-surgical treatments for post-burn scars: a network meta-analysis

Thank you again to the authors for their respectful and considered responses to the last review. I think your methods demonstrate a potential option for all future reviews to be made more interpretable and place greater weight on less popular (less cited) works.

Apologies, I have noticed two text errors which will need to be addressed in the Abstract please:

1. Define (or remove) the acronym SUCRA.

2. To rebalance the word count – remove sentence about risk of bias, it is not necessary.

Other than that, this reviewer believes that this version is of publishable standard. Well done!

7. PLOS authors have the option to publish the peer review history of their article (what does this mean? ). If published, this will include your full peer review and any attached files.

**Do you want your identity to be public for this peer review?** For information about this choice, including consent withdrawal, please see our Privacy Policy .

Reviewer #2: **Yes: ** Dale W Edgar

---

## [Author Response · Author response to Decision Letter 4]

28 Jul 2025

Dear Editors and Reviewers,

Thank you very much for giving us opportunities to revise our manuscript, and we appreciate the reviewer a lot for his positive and constructive comments and suggestions. We have studied reviewer’s comments carefully and have made revisions. We hope the corrections will meet with your approval.

Journal Requirement 1:

If the reviewer comments include a recommendation to cite specific previously published works, please review and evaluate these publications to determine whether they are relevant and should be cited. There is no requirement to cite these works unless the editor has indicated otherwise. Please review your reference list to ensure that it is complete and correct. If you have cited papers that have been retracted, please include the rationale for doing so in the manuscript text, or remove these references and replace them with relevant current references. Any changes to the reference list should be mentioned in the rebuttal letter that accompanies your revised manuscript. If you need to cite a retracted article, indicate the article’s retracted status in the References list and also include a citation and full reference for the retraction notice.

Response: We would like to thank the journal for the guidelines regarding the reference list. In response to the instructions, we have carefully reviewed all references cited in our manuscript to ensure completeness, accuracy, and relevance. We evaluated any references suggested by the reviewers and have included those that are pertinent to our study in the revised manuscript. Additionally, we have verified that none of the cited works have been retracted. All references are up-to-date, relevant, and appropriately formatted according to the journal’s citation style.

Thank you again for this thoughtful comments.

Editor :

Comment 1: We note that the grant information you provided in the “Funding Information” and “Financial Disclosure” sections do not match. When you resubmit, please ensure that you provide the correct grant numbers for the awards you received for your study in the “Funding Information” section.

Response: We sincerely thank the editor for the comments and for handling our manuscript. We have updated the “Funding Information” section to ensure that it contains the correct grant numbers, as shown below:

Find a Funder: Sichuan Province Science and Technology Program.

Award Number: 23ZDYF1907.

Grant Recipient: Not applicable.

Comment 2: Please note that funding information should not appear in the Acknowledgments section or other areas of your manuscript. We will only publish funding information present in the Funding Statement section of the online submission form. Please remove any funding-related text from the manuscript.

Response: Thanks for your helpful comment. We have carefully removed all funding-related information from the Acknowledgments section and any other parts of the manuscript, as requested. The funding details are now only provided in the Funding Information section of the online submission form, in accordance with the journal’s policy.

Reviewer 2:

Comment 1: Define (or remove) the acronym SUCRA.

Response: Thank you for your valuable comment. As suggested, we have added the full term “surface under the cumulative ranking curve” at its first mention in the Abstract section, as shown below:

A total of 17 studies and 1,013 participants were included in this analysis. The treatment ranking revealed that massage therapy demonstrated the most significant efficacy in reducing Vancouver Scar Scale score (surface under the cumulative ranking curve [SUCRA] = 89.0%), CO2 laser therapy exhibited the highest efficacy in decreasing scar thickness (SUCRA=96.8%) , and extracorporeal shock wave therapy + routine treatment showed the most significant efficacy in reducing Visual Analogue Scale score (SUCRA=58.6%).

Comment 2: To rebalance the word count – remove sentence about risk of bias, it is not necessary.

Response: Thanks for your valuable comment. As suggested, we have removed the sentence regarding the risk of bias assessment to rebalance the word count. The revised abstract now is shown as follows:

PubMed, Web of Science, Cochrane Library, PEDro, and Embase were systematically searched for eligible randomized controlled trial studies, and the network meta-analysis was performed via a frequentist approach. The primary outcomes assessed were Vancouver Scar Scale score, scar thickness and Visual Analogue Scale score.

---

## [Decision Letter · Decision Letter 4]

1 Aug 2025

Non-surgical treatments for post-burn scars: a network meta-analysis

PONE-D-24-50779R4

Dear Dr. Li,

We’re pleased to inform you that your manuscript has been judged scientifically suitable for publication and will be formally accepted for publication once it meets all outstanding technical requirements.

Kind regards,

Vincenzo Lionetti, M.D., PhD

Academic Editor

PLOS ONE

Additional Editor Comments (optional):

Reviewers' comments:

Reviewer's Responses to Questions

**Comments to the Author**

1. If the authors have adequately addressed your comments raised in a previous round of review and you feel that this manuscript is now acceptable for publication, you may indicate that here to bypass the “Comments to the Author” section, enter your conflict of interest statement in the “Confidential to Editor” section, and submit your "Accept" recommendation.

Reviewer #2: All comments have been addressed

2. Is the manuscript technically sound, and do the data support the conclusions?

Reviewer #2: Yes

3. Has the statistical analysis been performed appropriately and rigorously? 

Reviewer #2: Yes

4. Have the authors made all data underlying the findings in their manuscript fully available?

Reviewer #2: Yes

5. Is the manuscript presented in an intelligible fashion and written in standard English?

Reviewer #2: Yes

6. Review Comments to the Author

Reviewer #2: Thank you to the authors, all comments addressed. This reviewer happy to accept and believes that the manuscript is ready for publication if the editor agrees.

7. PLOS authors have the option to publish the peer review history of their article (what does this mean? ). If published, this will include your full peer review and any attached files.

**Do you want your identity to be public for this peer review?** For information about this choice, including consent withdrawal, please see our Privacy Policy .

Reviewer #2: **Yes: ** Dale W Edgar

---

## [Editor Report · Acceptance letter]

PONE-D-24-50779R4

PLOS ONE

Dear Dr. Li,

I'm pleased to inform you that your manuscript has been deemed suitable for publication in PLOS ONE. Congratulations! Your manuscript is now being handed over to our production team.

Kind regards,

on behalf of

Prof. Vincenzo Lionetti

Academic Editor

PLOS ONE